# DISCEDIT: Model Editing by Identifying Discriminative Components

**Chaitanya Murti**
Robert Bosch Centre for Cyberphysical Systems
Indian Institute of Science
mchaitanya@iisc.ac.in

**Chiranjib Bhattacharyya**
Computer Science and Automation
Robert Bosch Centre for Cyberphysical Systems
Indian Institute of Science
chiru@iisc.ac.in

## Abstract

Model editing is a growing area of research that is particularly valuable in contexts where modifying key model components, like neurons or filters, can significantly impact the model's performance. The key challenge lies in identifying important components useful to the model's predictions. We apply model editing to address two active areas of research, Structured Pruning, and Selective Class Forgetting. In this work, we adopt a distributional approach to the problem of identifying important components, leveraging the recently proposed *discriminative filters hypothesis*, which states that well-trained (convolutional) models possess discriminative filters that are essential to prediction. To do so, we define discriminative ability in terms of the Bayes error rate associated with the feature distributions, which is equivalent to computing the Total Variation (TV) distance between the distributions. However, computing the TV distance is intractable, motivating us to derive novel witness function-based lower bounds on the TV distance that require no assumptions on the underlying distributions; using this bound generalizes prior work such as Murti et al. [39] that relied on unrealistic Gaussianity assumptions on the feature distributions. With these bounds, we are able to discover critical subnetworks responsible for classwise predictions, and derive DISCEDIT-SP and DISCEDIT-U, algorithms for structured pruning requiring no access to the training data and loss function, and selective forgetting respectively. We apply DISCEDIT-U to selective class forgetting on models trained on CIFAR10 and CIFAR100, and we show that on average, we can reduce accuracy on a single class by over 80% with a minimal reduction in test accuracy on the remaining classes. Similarly, on Structured pruning problems, we obtain 40.8% sparsity on ResNet50 on Imagenet, with only a 2.6% drop in accuracy with minimal fine-tuning. [1]

## 1 Introduction

The black-box nature of neural networks makes understanding the precise mechanism by which a neural network makes a prediction (in the classification or regression settings), or generates a sample (in the generative settings) an active area of research [44], and relevant to several related problems,

---

[1]Our code is available at: `https://github.com/chaimurti/DisCEdit`

38th Conference on Neural Information Processing Systems (NeurIPS 2024).

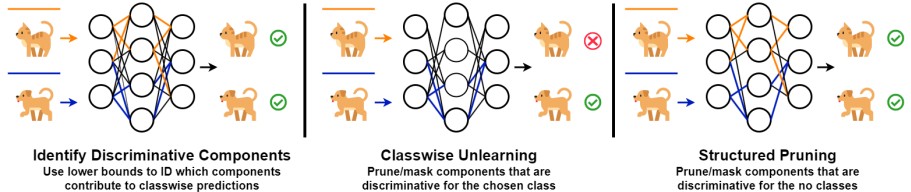

**Figure 1:** Identifying Discriminative Components for Model Unlearning and Structured Pruning

such as robustness, sparsity, and memorization. In particular, recent work aims to address this problem by identifying which *components* - such as neurons, convolutional filters, or attention heads - in the model contribute most significantly to predictions [44]. Prior work addressing this problem includes [2], which identified important filters by using human-defined concepts, and aligning them with filter responses to those concepts, whereas more recent work [48], uses an exhaustive regression-based approach. The field of structured pruning [19, 4] also provides a variety of heuristics for component importance, including first- and second-order derivative information [38, 37, 43], feature map ranks [35, 52], and layerwise reconstruction errors [59, 41]. Component attribution has also found attention in the fields of machine unlearning [58], where class-wise important components can be identified and edited to remove information about a given class or group[48, 23, 33].

In this work, we address this problem from a *distributional* milieu: that is, by analyzing the distributions of feature maps for models trained on datasets with distinct classes or subgroups. Specifically, we use the *discriminative filters hypothesis (DFH)* stated in [39], which states that well-trained models for classification possess a mix of *discriminative filters* - filters that yield feature maps with distinct class-conditional distributions - and non-discriminative filters and that the discriminative filters are important to model predictions. Discriminative filters can be used to identify which filters contribute to the prediction for samples from a given class. Under restrictive assumptions of Gaussianity, the DFH was used in [39] to derive a structured pruning algorithm that required only some form of distributional access, and no access to the loss function and training set. However, several key challenges remain. First, is it possible to identify discriminative filters without restrictive assumptions on the filter outputs? Second, can discriminative filters be used in settings other than structured pruning? Last, can the performance of algorithms leveraging the DFH be improved upon with better identification of discriminative filters? In this work, we answer each of the questions affirmatively.

First, to identify discriminative filters without unrealistic assumptions on the feature map distributions, we derive tractable, witness function-based lower bounds on the Total Variation (TV) distance, with which we also reveal hitherto unknown ties between the TV distance and classical, discrimination-based classifiers. We then use our proposed methodology to identify discriminative filters for classwise model unlearning and structured pruning, which we call DISCEDIT: **Dis**criminative **C**omponent identification for model **Edit**ing. Our methodology is simple - we first identify discriminative filters for a given class, and prune them (for class forgetting), called DISCEDIT-U, or in the case of structured pruning, prune filters that are non-discriminative for all classes, called DISCEDIT-SP. Note that our method for identifying discriminative filters requires only distributional access, and neither the training data nor the loss function. We illustrate our approach in Figure 1, and formally state our contributions below.

**Discriminative ability of a filter:** We quantify the discriminative ability of a filter in a neural network, as the worst possible Bayes error rate of binary classifiers trained upon the features it generates(See Section 3). Since, in general, computing the Bayes' error rate is intractable, we seek to approximate it using lower bounds on the TV distance.

**Witness function based Distribution Agnostic Lower bound on TV distance:** In order to identify discriminative filters without distributional assumptions, we propose a novel witness function-based lower bound on the TV distance between distributions to address this gap in Theorem 1. We propose a lower bound that relies only on knowledge of finitely many moments(Theorem 2) and derive another lower bound accounting for moment measurement errors in Theorem 3. These bounds do not require distribution-specific assumptions and hence enable this work to generalize previous work [39], which requires the distributions to be Gaussian( which is an unrealistic assumption). To be noted that these bounds are of independent interest as they are broadly applicable. Moreover, using a careful choice of witness function, our bounds reveal new connections between discriminant-based classifiers like

the Fisher linear discriminant and the Minimax Probability Machine, and the TV distance, which we state in Corollary 2.

**Model Editing using Lower Bounds:** We apply our lower bounds to the problem of identifying which components in a model contribute to predictions from certain classes.

1. **Class Unlearning:** We identify components capable of discriminating each class, which we then prune in order to unlearn that class, requiring no access to the loss function, called DISCEDIT-U.

2. **Pruning without the training data or loss function:** Next, we derive a family of algorithms for structured pruning requiring no access to the training data or loss function by identifying non-discriminative filters with our proposed lower bounds, called DISCEDIT-SP.

**Experimental Validation:** We produce a slate of experiments to illustrate the efficacy of identifying discriminative components for structured pruning and machine unlearning/class forgetting. We compare the efficacy of DISCEDIT-SP for pruning VGG nets and ResNets trained on CIFAR10, and show that we achieve up to 76% sparsity with a .12% reduction in accuracy on VGG19, and on Imagenet, we achieve 40.8% sparsity with a 2.58% drop in accuracy with minimal fine-tuning. Similarly, for machine unlearning, we show that our method enables 83% drop in accuracy on the class to be forgotten, with a 1.2% increase in accuracy on the remaining classes, without any finetuning.

## 1.1 Related Work

In this section, we introduce a brief summary of related literature. We provide a more detailed discussion of the literature in Appendix A.

**Structured pruning** Structured pruning reduces the inference time and power consumption of neural networks by removing entire filters or neurons, as opposed to unstructured pruning that removes individual weights [29, 18]. A variety of structured pruning methods require the loss function, and identify import components using gradient information [38, 30], or approximations of the Hessian of [29, 18, 51, 36].

A variety of structured pruning methods that do not rely on the loss function have been proposed, which identify important components by norms of the weights [31, 32], bounds on the reconstruction error incurred by pruning a filter [41, 59], the rank of the feature maps [35, 52], and coresets of feature maps [34]. More recent approaches identify components based on the discriminative ability of the corresponding featuremaps, using metrics like the Hellinger distance between class-conditional feature maps [39], or Fisher discriminant-based methods [20, 21]. In this work, we formalize the notion of discriminative ability in terms of the Bayes risk and derive novel lower bounds on the total variation distance to approximate it effectively. Unlike previous works, we make no assumptions about the class-conditional distributions, and our pruning algorithms require no access to the training data or loss function. For a more comprehensive discussion on structured pruning, we refer readers to Appendix A or the surveys [19, 4].

**Machine unlearning** Machine Unlearning has recently received significant attention due to concerns like data privacy and security [5, 42]. A variety of works propose methods to forget data points while maintaining model accuracy, even in adaptive settings [47, 16, 22, 14]. Selective forgetting, where classes or groups are forgotten, connects machine unlearning to continual learning [58, 57]. Model editing for unlearning, however, remains an underexplored area of research. Recent studies machine unlearning can be enhanced by sparsifying models [23, 46], and discrimination-aware pruning has been used in federated settings [56] and for forgetting specific classes [48]. Our work differs by directly utilizing the discriminative ability of model components to identify and remove those responsible for a given class, enabling effective unlearning without access to the original training data. For a more detailed discussion, we refer readers to Appendix A.

## 2 Background and Notation

In this section, we introduce our notation and provide basic background definitions.

**Notation**  For an integer $N > 0$, let $[N] := \{1, \cdots, N\}$. Let $\mathbf{0}_N$ be a vector of zeros of dimension $N$. Let $\text{sort}_B(\{a_1, \cdots, a_M\})$ be the set of the $B$ largest elements of $\{a_1, \cdots, a_M\}$. Suppose $\mathbb{P}$, $\mathbb{Q}$ are two distributions supported on $\mathcal{X}$, with densities given by $p(x)$ and $q(x)$. For a function $f : \mathcal{X} \to \mathbb{R}$, let $\bar{f}_p = \mathbb{E}_{x \sim \mathbb{P}}[f(x)]$, and let $\bar{f}_p^{(2)} = \mathbb{E}_{x \sim \mathbb{P}}[f(x)^2]$. Let $\mathcal{D}$ be a data distribution. Suppose the dataset has $C$ classes, then let $\mathcal{D}_c$ be the class-conditional distribution of the $c$-th class, and let $\mathcal{D}_{\bar{c}}$ be the distribution of the complement of $c$ (that is, samples are drawn from all classes other than $c$).

Suppose we have a neural network $\mathcal{W} = (W_1, \cdots, W_L)$. Each layer yields (flattened) representations

$$Y^l(x) = \left[ Y_1^l(X), \cdots, Y_{N_l}^l(X) \right], \tag{1}$$

where $N_l$ is the number of filters in layer $l$. Since $Y^l$ is dependent on $X$, we assume that $Y^l(X) \sim \mathcal{D}^l$, and $Y_j^l(X) \sim \mathcal{D}_j^l$. Furthermore, let $\mathcal{D}_{j,c}^l$ and $\mathcal{D}_{j,\bar{c}}^l$ be the class-conditional distributions and class-complement distributions of $Y_j^l(X)$ respectively.

**Background**  In this section, we provide relevant background for this work. First, let $\mathbb{P}$ and $\mathbb{Q}$ be distributions supported on $\mathcal{X}$, with moments $\mu_p, \Sigma_p$ and $\mu_q, \Sigma_q$. Then, recall that

$$\text{Fish}(\mathbb{P}, \mathbb{Q}; u) = \frac{\left( u^\top (\mu_p - \mu_q) \right)^2}{u^\top (\Sigma_p + \Sigma_q) u} \quad \text{and} \quad \text{MPM}(\mathbb{P}, \mathbb{Q}; u) = \frac{|u^\top (\mu_p - \mu_q)|}{\sqrt{u^\top \Sigma_p u} + \sqrt{u^\top \Sigma_q u}}. \tag{2}$$

where $\text{Fish}$ denotes the Fisher discriminant [24], and $\text{MPM}$ denotes the Minimax probability machine [27, 28]. If we choose the optimal $u$, denoted by $u^*$, we write $\max_u \text{Fish}(\mathbb{P}, \mathbb{Q}; u) = \text{Fish}(\mathbb{P}, \mathbb{Q}; u^*) = \text{Fish}^*(\mathbb{P}, \mathbb{Q})$ and $\max_u \text{MPM}(\mathbb{P}, \mathbb{Q}; u) = \text{MPM}(\mathbb{P}, \mathbb{Q}; u^*) = \text{MPM}(\mathbb{P}, \mathbb{Q})^*$.

We define the TV and Hellinger distances as follows.

**Definition 1.** *Let $\mathbb{P}$ and $\mathbb{Q}$ be two probability measures supported on $X$, and let $p$ and $q$ be the corresponding densities. Then, we define the Total Variation Distance* TV *as*

$$\text{TV}(\mathbb{P}, \mathbb{Q}) = \sup_{A \subset X} |\mathbb{P}(A) - \mathbb{Q}(A)| = \frac{1}{2} \int_X |p(x) - q(x)| \, dx$$

The *Bayes Optimal classifier*, as given in [9, 8], associated with distributions $\mathbb{P}$ and $\mathbb{Q}$ with labels $-1$ and $1$ respectively is given by

$$f(x) = \arg\max_{c \in \{-1, 1\}} \Pr(c|x),$$

and has the error rate $R^\star(\mathbb{P}, \mathbb{Q})$, called the *Bayes Error Rate*. Next, we relate the Bayes classifier and the Bayes error rate (as described in, say, Devroye et al. [8]) of a classifier to the TV distance with the following identity. Consider a binary classification problem with instance $x$ and labels $c \in \{-1, 1\}$, with class conditional distributions given by $\mathbb{P}$ and $\mathbb{Q}$. The Bayes error rate satisfies the identity

$$2R^*(\mathbb{P}, \mathbb{Q}) = 1 - \text{TV}(\mathbb{P}, \mathbb{Q}). \tag{3}$$

# 3   Editing Models by Identifying Discriminative Components

There are three central questions addressed in this work:

1. **Model Unlearning:** How do we edit components in order to reduce accuracy on certain groups or classes only?

2. **Structured pruning:** How do we remove components to ensure that the accuracy of all classes is minimally affected?

3. **Determining the Discrimination Ability of a Model Component:** How do we assess the Discrimination ability of filters without access to the training data or loss function, and without making any assumptions about the class-conditional feature distributions?

Inspired by Murti et al. [39], the key idea in this work for addressing each of the questions raised above is to identify *discriminative components*, that yield feature maps upon which accurate classifiers can be trained, and thus with distinct class-conditional distributions. A heuristic to address this

problem would be to train a classifier upon the feature map; those features upon which accurate classifiers can be trained are generated by discriminative filters; however, this is highly impractical. Thus, as with [39], we address this problem by identifying filters for which the class-conditional distributions of the feature maps are distinct, which are identified based on estimates of the total variation distance between the class-conditional distributions of the associated feature maps. In this work, we formalize *discriminative ability* of a component in terms of the best possible classifier that can be trained on the features generated by it. In the sequel, we illustrate similar but distinct notions of discriminative ability important for classwise unlearning, and structured pruning. We first formally define *discriminative ability* and the *class-c discriminative ability* as follows.

**Definition 2** (Discriminative Ability). *Consider a CNN with $L$ layers trained on a dataset with $C$ classes, and consider the $j$th filter in the $l$th layer. The **class-$c$ discriminative ability** $\eta_{l,j}^c$, and the **discriminative ability** $\eta_{l,j}$ of the filter are given by*

$$\eta_{l,j}^c = R^* \left( \mathcal{D}_{j,c}^l, \mathcal{D}_{j,\bar{c}}^l \right) \qquad\qquad \eta_{l,j} = \max_{c \in [C]} \ \eta_{l,j}^c = \max_{c \in [C]} \ R^* \left( \mathcal{D}_{j,c}^l, \mathcal{D}_{j,\bar{c}}^l \right). \qquad (4)$$

**Classwise Unlearning** For classwise unlearning, we aim to identify those components that are responsible for predictions of a selected class or group within the dataset. Thus, we aim to identify *those components that are able to discriminate between the given class, say with label c, and others.* As with Wang et al. [58], we say $\mathcal{D}_c$ is the **Forget Set**, and $\mathcal{D}_{\bar{c}}$ is the **Remain Set**. As noted in Shah et al. [48], Jia et al. [23], our aim is to minimize the test accuracy of the model on $\mathcal{D}_c$, while maintaining the test accuracy on $\mathcal{D}_{\bar{c}}$. Thus, to unlearn class $c$, we edit those components for which the class-$c$ discriminative ability, $\eta_{l,j}^c$ is low. An interesting point to note is that some components may be discriminative only for class $c$, which we call *class-selective components*, whereas others may be discriminative for multiple classes. Note that when the dataset has a large number of classes, our experiments indicate that class-specific components generally can't be found.

**Structured Pruning:** Identifying discriminative components (specifically filters in convolutional networks) for structured pruning was first introduced in Murti et al. [39]. Structured pruning involves removing components for a network while maintaining the accuracy of the classifier. Following Murti et al. [39], we aim to *retain components that are discriminative for multiple classes*. Our goal is to remove components from the model while maintaining the model's accuracy on each class conditional $\mathcal{D}_c$. Thus, Definition 2 gives us a means by which we can identify discriminative components based on the worst Bayes error rate for discriminating the class conditional distributions of the given feature map. Filters for which $\eta_{l,j}$ is small are considered discriminative, as the best possible classifier that can be trained on those features will be highly accurate.

**Assessing the Discriminative Ability:** In general, the Bayes error rate cannot be computed. However, since the TV distance and the Bayes error rate are connected by the identity (3), which states that for two distributions $\mathbb{P}, \mathbb{Q}$, the Bayes risk is given by $\frac{1}{2}(1 - \mathtt{TV}(\mathbb{P}, \mathbb{Q}))$, we can reformulate our distributional pruning strategy as identifying those filters that generate features for which class-conditional distributions are well-separated in the Total Variation sense, and prune them. In [39], this was achieved by making the strong assumption that the distributions of the class-conditional feature maps were spherical Gaussian. In the sequel, we propose novel lower bounds on the total variation distance that require no restrictive Gaussianity assumptions, thereby enabling us to effectively identify discriminative components.

## 4 Witness Function-Based Lower Bounds for the Total Variation Distance

In this section, we derive lower bounds on the Total Variation Distance that rely on the moments of a *witness function*, a scalar-valued function whose moments can be used to derive bounds on divergences between distributions. More formally, we write $f : \mathcal{X} \to \mathbb{R}$, where $\mathcal{X}$ is the support of the distribution(s) in question. We then adapt our lower bound to a variety of scenarios, depending on the extent of the information about the distributions available to us. When access to only the first two moments is available, we derive lower bounds on the total variation distance based on the Fisher linear discriminant and the minimax probability machine.

Estimating the Total Variation distance is known to be #P complete [3]. Estimating lower bounds on the TV distance is an active area of research (see Davies et al. [7] and the references within), with a variety of applications from clustering [1, 17] to analyzing neural networks [59]. However, most

bounds such as those presented in Davies et al. [7] require prior knowledge about the distributions, and tractable estimation of lower bounds given access to collections of moments or samples, without assumptions on the distributions themselves, remains an open problem.

## 4.1 Witness Function-based Lower Bounds on the TV Distance

In this section, we propose lower bounds on the TV distance that only requires access to the moments of a *witness function*, as described in Gretton et al. [15], Kübler et al. [25].

**Theorem 1.** *Let $\mathbb{P}, \mathbb{Q}$ be two probability measures supported on $\mathcal{X} \subseteq \mathbb{R}^d$, and let $p$ and $q$ be the corresponding densities. Let $\mathcal{F}$ be the set of functions with bounded first and second moments defined on $\mathcal{X}$. Then,*

$$\text{TV}(\mathbb{P}, \mathbb{Q}) \geq \sup_{f \in \mathcal{F}} \frac{\left(\bar{f}_p - \bar{f}_q\right)^2}{2\left(\bar{f}_p^{(2)} + \bar{f}_q^{(2)}\right)} \tag{5}$$

*Proof Sketch.* We provide a sketch of the proof. We express the quantity $f_p - f_q$ in terms of the densities $p(X)$ and $q(X)$. We then isolate the integral of $|p(x) - q(x)|$. After rearranging terms, we obtain the result. For the full proof, we refer readers to Appendix B. $\qquad\square$

## 4.2 Moment-based Lower Bounds on the TV distance

Motivated by the need to identify discriminative filters when we only have access to the moments of feature map distributions, we propose worst-case lower bounds on the TV distance given access to finitely many moments of the distributions $\mathbb{P}$ and $\mathbb{Q}$.

Let $\mathcal{S}_k(\mathbf{P}) := \{\mathbb{P} : \mathbb{E}_{X \sim \mathbb{P}}[X_1^{d_1} \cdots X_n^{d_n}] = \mathbf{P}_{d_1 \cdots d_n}, \sum_i d_i \leq k\}$ be the set of probability measures whose moments are given by $\mathbf{P}$, where $\mathbb{E}_{X \sim \mathbb{P}}\left[X_1^{d_1} \cdots X_n^{d_n}\right] = \mathbf{P}_{d_1 \cdots d_n}$; similarly, let $\mathcal{S}_k(\mathbf{Q})$ be the set of measures whose moments are given by $\mathbf{Q}$. For any random variable $X \in \mathbb{R}^d$ supported on $\mathcal{X}$, suppose $\varphi : \mathbb{R}^d \to \mathbb{R}^n$ for which there exist functions $g$ and $G$ such that $\mathbb{E}_X[\varphi(X)] = g(\mathbf{P})$ and $\mathbb{E}\left[\varphi(X)\varphi(X)^\top\right] = G(\mathbf{P})$. Given two collections of moments of the same order, we want to measure the worst-case TV separation between *all* distributions possessing the moments given in $\mathbf{P}$ and $\mathbf{Q}$. We define this as

$$D_{\text{TV}}(\mathcal{S}_k(\mathbf{P}), \mathcal{S}_k(\mathbf{Q})) = \min_{\mathbb{P} \in \mathcal{S}_k(\mathbf{P}), \mathbb{Q} \in \mathcal{S}_k(\mathbf{Q})} \text{TV}(\mathbb{P}, \mathbb{Q}) \tag{6}$$

For the sake of brevity, we write $D_{\text{TV}}(\mathcal{S}_k(\mathbf{P}), \mathcal{S}_k(\mathbf{Q})) = D_{\text{TV}}(\mathbf{P}, \mathbf{Q}; k)$.

**Theorem 2.** *Suppose $\mathbf{P}$ and $\mathbf{Q}$ are sets of moments of two probability measures supported on $\mathcal{X}$. Let $\varphi(X)$ be a vector of polynomials such that $\mathbb{E}_{\mathbb{P}}[\varphi(X)] = g(\mathbf{P})$, $\mathbb{E}_{\mathbb{Q}}[\varphi(X)] = g(\mathbf{Q})$, $\mathbb{E}_{\mathbb{P}}[\varphi(X)\varphi(X)^\top] = G(\mathbf{P})$, and $\mathbb{E}_{\mathbb{Q}}[\varphi(X)\varphi(X)^\top] = G(\mathbf{Q})$, and let $f = u^\top(\varphi(X) - \frac{g(\mathbf{P}) + g(\mathbf{Q})}{2})$, be a witness function. Then, for any $\mathbb{P} \in \mathcal{S}_k(\mathbf{P})$, $\mathbb{Q} \in \mathcal{S}_k(\mathbf{Q})$, supported on a set $\mathcal{X} \subseteq \mathbb{R}^d$, we have*

$$D_{\text{TV}}(\mathbf{P}, \mathbf{Q}; k) \geq S_{\text{TV}}^*(\mathbf{P}, \mathbf{Q})\left(2 + S_{\text{TV}}^*(\mathbf{P}, \mathbf{Q})\right)^{-1} \geq S_{\text{H}}^*(\mathbf{P}, \mathbf{Q})^2\left(\sqrt{2} + S_{\text{H}}^*(\mathbf{P}, \mathbf{Q})\right)^{-2}, \text{ where:}$$

$$S_{\text{TV}}^*(\mathbf{P}, \mathbf{Q}) = (\Delta g)^\top(\tilde{G}(\mathbf{P}) + \tilde{G}(\mathbf{Q}))^{-1}(\Delta g) \text{ and } S_{\text{H}}^*(\mathbf{P}, \mathbf{Q}) = \max_u \frac{|u^\top(g(\mathbf{P}) - g(\mathbf{Q}))|}{\sqrt{u^\top G(\mathbf{P})u} + \sqrt{u^\top G(\mathbf{Q})u}}$$

*and $\Delta g = g(\mathbf{P}) - g(\mathbf{Q})$ and $\tilde{G}(\mathbf{P}) = G(\mathbf{P}) - g(\mathbf{P})g(\mathbf{P})^\top$.*

*Proof Sketch.* We apply Theorem 1 with the given witness function, and obtain an expression in terms of $S_{\text{TV}}^*(\mathbf{P}, \mathbf{Q})$. Since the bound holds for any distributions that yield the given moments of the witness function, the statement holds. The full proof is provided in Appendix B. $\qquad\square$

Theorem 2 is a worst-case lower bound on the TV distance between distributions with given truncated moment sequences. While we focus our results on the case where $f(x) = u^\top \varphi(x)$, where $\varphi(x)$ is a vector of monomials, this bound is valid for any choice of $f$ with bounded first and second moments.

### 4.3 Computing $\mathtt{TV}(\mathbb{P}, \mathbb{Q})$ from the Lower Bound

In general, the bounds proposed in Theorem 1 are not tight. However, an interesting observation is that the lower bound proposed in Theorem 1 can, in certain cases, be used to compute the Bayes optimal classifier, and thus the true TV distance. Specifically, if the Bayes optimal classifier lies in a given set of functions $\mathcal{F}$, the bound can be used to compute the Bayes classifier. We state one such case below in Corollary 1.

**Corollary 1.** *Suppose* $\mathbb{P} \equiv \mathcal{N}(\mu_p, \Sigma)$ *and* $\mathbb{Q} \equiv \mathcal{N}(\mu_q, \Sigma)$ *Let* $f(x; u) = u^\top (x - \frac{1}{2}(\mu_p - \mu_q))$ *be a witness function. Then,*

$$u^\star = \arg\max_u \ \frac{(\mathbb{E}_{x \sim \mathbb{P}}[f(x; u)] - \mathbb{E}_{x \sim \mathbb{Q}}[f(x; u)])^2}{\mathbb{E}_{x \sim \mathbb{P}}[f(x; u)^2] + \mathbb{E}_{x \sim \mathbb{Q}}[f(x; u)^2]} = \Sigma^{-1}(\mu_p - \mu_q)$$

*is the weight vector of the Bayes optimal classifier* $f_{Bayes}(x) = \text{sign}\left(u^{\star\top} x + b\right)$*, and* $\mathtt{TV}(\mathbb{P}, \mathbb{Q}) = 2\Phi\left(\sqrt{(u^\star)^\top(\mu_p - \mu_q)}/2\right) - 1$.

*Proof Sketch.* We find the $u^\star$ that maximizes the the TV lower bound given in Theorem 1, and then apply the same to the exact expression of the TV distance between Gaussian measures with the same covariance. The full proof is provided in Appendix B. $\qquad\square$

*Remark* 4.1. This result illustrates the case where the Bayes' classifier lies in the set of functions $\mathcal{F} := \{f(x) : f(x) = u^\top \varphi(x)\}$ for a given function $\varphi(x)$. In this case, if $\varphi(x) = x - \frac{1}{2}(\mu_p - \mu_q)$, and $\mathbb{P}$ and $\mathbb{Q}$ are Gaussian with the same variance, the Bayes classifier is equivalent to the Fisher discriminant.

### 4.4 Connections to Discriminant Based Classifiers

In this section, we exploit the bound stated in Theorem 1 to reveal extensive connections between the total variation distance and discriminant-based linear classifiers, specifically the Fisher Linear Discriminant and the Minimax Probability Machine, that are of independent interest to readers.

Specifically, we show that the TV distance is lower-bounded by monotonic functions of the Fisher Discriminant and the Minimax Probability Machine. We state this result formally in Corollary 2.

**Corollary 2.** *Let* $\mathbb{P}, \mathbb{Q}$ *be two probability measures supported on* $X \subseteq \mathbb{R}^d$*, let* $p$ *and* $q$ *be the corresponding densities, and let* $\mu_p$*,* $\mu_q$ *and* $\Sigma_p$*,* $\Sigma_q$ *be the means and variances of* $\mathbb{P}$ *and* $\mathbb{Q}$ *respectively. Then,*

$$\mathtt{TV}(\mathbb{P}, \mathbb{Q}) \geq \frac{\mathsf{Fish}^*(\mathbb{P}, \mathbb{Q})}{2 + \mathsf{Fish}^*(\mathbb{P}, \mathbb{Q})} \geq \left(\frac{\mathsf{MPM}^*(\mathbb{P}, \mathbb{Q})}{\sqrt{2} + \mathsf{MPM}^*(\mathbb{P}, \mathbb{Q})}\right)^2.$$

*Proof.* Choose $\varphi(x) = x$. Thus, $g(\mathbf{P}) = \mu_p$, and $G(\mathbf{P}) = \Sigma_p + \mu_p \mu_p^\top$. Substitute these into Theorem 2 to complete the proof. $\qquad\square$

This lower bound can be improved upon by selecting a witness function of the form $f(x; u) = u^\top \varphi(x)$ where $\varphi(x)$ is a vector of basis functions (such as monomials, if $f(x; u)$ is a polynomial). Moreover, lower bounds that are robust to estimation error, based on the Fisher Discriminant and Minimax Probability Machines, can be derived by directly applying the techniques proposed in [24] (for the Fisher discriminant case) and [28] (for the Minimax Probability Machine case). We discuss this in Section 6.2.

### 4.5 Lower bounds Robust to Estimation Errors

The lower bounds derived in Theorems 1 and 2 are functions of moments of the distributions, which must typically be estimated from samples. In practice, we use plug-in estimators for $g(\mathbf{P})$, $g(\mathbf{Q})$, $G(\mathbf{P})$ and $G(\mathbf{Q})$. Since these estimators are computed using samples, errors in estimation arise, which in turn creates errors in computing the lower bound.

This estimation error is a particularly challenging issue in high dimensions, where 'high dimensions' refers to the setting where the number of samples $n$ is significantly less than the dimension of the data, $d$. However, in typical neural networks such as VGG-nets and ResNets, the feature maps, particularly of the layers close to the output that can be effectively pruned (see, for instance, [34, 39]), have low dimensional feature maps. For instance, on VGG16 trained on CIFAR10, the feature maps generated by the 5th layer are of dimension 64; thus, for a relatively small number of samples $n$, the dimension is less than $n$.

In this section, we present robust lower bounds for the case when $f(x) = u^\top x$, and the moments being estimated are $\mu_p = \mathbb{E}_\mathbb{P}[x]$ and $C_p = \mathbb{E}_\mathbb{P}[xx^\top]$ using plug-in estimators of the form

$$\bar{\mu}_p = \frac{1}{N}\sum_{i=1}^N x_i \quad \text{and} \quad \bar{C}_p = \frac{1}{N}\sum_{i=1}^N x_i x_i^\top \tag{7}$$

where $x_i$ are drawn iid from $\mathbb{P}$. We assume that we can quantify the estimation error for the above moments, and can apply lower bounds as proposed in Kim et al. [24] accordingly.

**Theorem 3.** *Suppose $\mathbb{P}, \mathbb{Q}$ be two probability measures supported on $X \subseteq \mathbb{R}^d$, with densities $p$ and $q$, and let $\mu_p = \mathbb{E}_\mathbb{P}[x]$, $\mu_q = \mathbb{E}_\mathbb{Q}[x]$ and $C_p = \mathbb{E}_\mathbb{P}[xx^\top]$, $C_q = \mathbb{E}_\mathbb{Q}[xx^\top]$. Suppose we have plug-in estimates $\bar{\mu}_p, \bar{C}_p, \bar{\mu}_q, \bar{C}_q$ as defined in (7), that satisfy*

$$\|\mu_p - \bar{\mu}_p\|_2 \le \delta_p \text{ and } \|\mu_q - \bar{\mu}_q\|_2 \le \delta_q. \ \|C_p - \bar{C}_p\|_F \le \rho_p \text{ and } \|C_q - \bar{C}_q\|_F \le \rho_q.$$

*Then, with a witness function of the form $f(x) = u^\top x$*

$$D_{\mathrm{TV}}(\mathbb{P}, \mathbb{Q}; 2) \ge \min_{\mu_p, \mu_q \in \mathcal{M}} (\Delta\mu)^\top (C_p + C_q + \rho I)^{-1}(\Delta\mu), \tag{8}$$

*where $\mathcal{M} = \{(\mu_p, \mu_q) : \|\mu_p - \bar{\mu}_p\|_2 \le \delta_p, \|\mu_q - \bar{\mu}_q\|_2 \le \delta_q\}$, $\Delta\mu = \mu_p - \mu_q$, and $\rho = \rho_p + \rho_q$.*

*Proof Sketch.* The proof is similar to the derivation of (15) in Kim et al. [24]. A full proof is provided in Appendix B $\qquad\square$

This result can also be applied to the estimation error of functions of $x$, such as a vector of monomials $\varphi(x)$, provided the estimation error of each moment can be bounded.

## 5 DISCEDIT: Distributional Algorithms for Model Editing

In this section, we leverage the lower bounds proposed in Theorems 1 and 2, and Corollary 2 to develop one-shot algorithms for model editing that require no access to the training data or loss function, but only access to the data distributions. We propose two algorithms, DISCEDIT-SP and DISCEDIT-U, that identify discriminative components (filters in CNNs), and prunes them to either unlearn a class (DISCEDIT-U) or to sparsify the model with minimal loss of accuracy (DISCEDIT-SP). A variety of variants of these algorithms based on different witness functions are provided in Appendix D.

### 5.1 DISCEDIT-SP: A Distributional Approach to Structured Pruning

In this section, we propose DISCEDIT-SP, an algorithm for structured pruning that identifies filters (in convolutional networks) that are discriminative for multiple classes, and retains them. Unlike the approach proposed in Murti et al. [39], no restrictive assumptions on the Gaussianity of class-conditional feature distributions is needed. Furthermore, by assuming the distributions are Gaussian and using the closed-form Hellinger lower bound, $\binom{C}{2}$ pairwise TV distances need to be computed for each filter. We now state the DISCEDIT-SP algorithm.

Let $Y^l(X)$ be the features generated by layer $l$ of a neural network as defined in (1). We choose a witness function $f = u^\top \varphi(Y_j^l(X)) = \varphi_j^l(X)$, and let $\bar{f}_{l,j,c}(u) = \mathbb{E}_{X \sim \mathcal{D}_c}[u^\top \varphi_j^l(X)]$, $\bar{f}_{l,j,\bar{c}}(u) = \mathbb{E}_{X \sim \mathcal{D}_{\bar{c}}}[u^\top \varphi_j^l(X)]$, $\bar{f}_{l,j,c}^{(2)}(u) = \mathbb{E}_{X \sim \mathcal{D}_c}[(u^\top \varphi_j^l(X))^2]$ and $\bar{f}_{l,j,\bar{c}}^{(2)}(u) = \mathbb{E}_{X \sim \mathcal{D}_{\bar{c}}}[(u^\top \varphi_j^l(X))^2]$. Next, define $r_j^l$ to be the saliency score for the $j$th filter in the $l$th layer as

$$r_j^l = \min_{c \in [C]} \max_u \ (\bar{f}_{l,j,c}(u) - \bar{f}_{l,j,\bar{c}}(u))^2 \left(\bar{f}_{l,j,c}^{(2)}(u) + \bar{f}_{l,j,\bar{c}}^{(2)}(u)\right)^{-1}. \tag{9}$$

We use the lower bound established in Theorem 1 on the TV distances between the class conditional distributions to measure the *saliency* or importance of a given filter. With this, we formally state DISCEDIT-SP in Algorithm 1.

The DISCEDIT-SP algorithm has several advantages. First, as compared to TVSPrune, it requires that only $C$ TV distances be computed at each step. Second, by varying the choice of witness function, we obtain new algorithms for structured pruning; we can choose different witness functions for each class as well.

### 5.2 DISCEDIT-U:
### A Distributional Approach to Machine Unlearning

We now state an algorithm for classwise unlearning based on model editing, called DISCEDIT-U. Motivated by works such as Shah et al. [48], our algorithm requires identifying and editing (specifically pruning) class-selective components for the class which is to be unlearned. DISCEDIT-U uses the same setup as DISCEDIT-SP. However, DISCEDIT-U only requires identifying discriminative filters for a single class, we have

$$r_j^l = \max_u \left( \bar{f}_{l,j,c}(u) - \bar{f}_{l,j,\bar{c}}(u) \right)^2 \left( \bar{f}_{l,j,c}^{(2)}(u) + \bar{f}_{l,j,\bar{c}}^{(2)}(u) \right)^{-1}.$$
(10)

---

**Algorithm 1:** DISCEDIT-X

**Input:** Class conditional distributions $\mathcal{D}_c$ and class-complements $\mathcal{D}_{\bar{c}}$ for all $c \in [C]$, Pretrained CNN with parameters $\mathcal{W} = (W_1, \cdots, W_L)$, layerwise sparsity budgets $B^l$, witness function $f$

**for** $l \in [L]$ **do**
    Set $S^l = [s_1^l, \cdots, s_{N_l}^l] = \mathbf{0}_{N_l}$
    **if** X *is* SP **then**
        For each $j$, compute $r_j^l$ using (9).
        **if** $j \in \text{sort}_{B_l}(\{r_j^l\}_{j=1}^{N_l})$ **then**
            Set $s_j^l = 1$
    **if** X *is* U *and Forget Class is* c **then**
        For each $j$, compute $r_j^l$ using (10).
        **if** $j \in \text{sort}_{B_l}(\{r_j^l\}_{j=1}^{N_l})$ **then**
            Set $s_j^l = 0$

**Output:** Binary masks $S^1, \cdots, S^L$
**return** $\hat{\mathcal{W}}$

---

We formally state DISCEDIT-U in Algorithm 1. The DISCEDIT-U algorithm has several advantages. As we show in our experiments in Appendix E, few samples are required to effectively compute the witness functions.

## 6 Empirical Evaluations

In this section, we empirically study the effectiveness of identifying discriminative filters for model editing tasks, specifically structured pruning and class unlearning. Additional experimental details are given in Appendix E. Experiments showing that class-conditional feature map distributions are non-gaussian, the effectiveness of variants of witness functions, the effectiveness of sparsity in class forgetting, and other ablations, are provided in Appendix E. Our experiment setup is provided in Appendix F, along with baseline accuracies of all models, shown in Table 11.

### 6.1 Identifying Discriminative Subnetworks and Class Unlearning with DISCEDIT-U

In this section, we investigate the utility of the lower bounds given in Theorems 1-3 in identifying discriminative subnetworks and for class unlearning.
**Experiment Setup:** We investigate VGG16, Resnet56, ResNet20, and a custom ViT (details given in Appendix F) trained on CIFAR10, and VGG16 and ResNet56 models trained on CIFAR100,

---

**Table 1:** A summary of results of class unlearning using DISCEDIT-U. We take the average over the Forget and Remain accuracies (FA and RA) after applying DISCEDIT-U to each class. Note that FA=accuracy drop on forget class, RA=accuracy drop on remain set. Values are averaged over 10 trials. NoFT refers to using DISCEDIT-U without fine-tuning, and with pruning only 5.4% of weights for VGG16, 1.8% of weights for ResNet56, 1.8% of weights for ResNet20, whereas FT refers to using DISCEDIT-U with 1 epoch of fine-tuning, and with pruning ratios of 18.4%, 22%, 16.6%, and 10.2% for VGG16, ResNet56, ResNet20, and our ViT respectively .

| Dataset | Model | Our Work | | | | Baselines | | | |
|---|---|---|---|---|---|---|---|---|---|
| | | FA (NoFT) | RA (NoFT) | FA (FT) | RA (FT) | FA (GA) [23] | RA (GA) [23] | RA (IU) [23] | RA (IU) [23] |
| CIFAR-10 | VGG-16 | **8.7%** | 82.5% | **3.7%** | 91.6% | 22.5% | 88.8% | 11.42% | 89.8% |
| | ResNet56 | 16.3% | 85.9% | 9.7% | 89.4% | - | - | - | - |
| | ResNet20 | **9.4%** | 83.9% | **6.0%** | 86.6% | 11.52% | 85.46% | - | - |
| | ViT | 16.5% | 66.3% | 11.0% | 71.2% | - | - | - | - |
| CIFAR-100 | VGG16 | 11.3% | 68.0% | 10.7% | 72.9% | - | - | - | - |
| | ResNet56 | 31.1% | 60.4% | 17.9% | 68.7% | - | - | - | - |
| | ViT | 13.1% | 44.2% | 14.6% | 61.0%% | - | - | - | - |

and identify subnetworks responsible for predictions from each class. We use the witness function $f(x) = \exp(\|x\|^2)$ in our experiments, using 256 samples from the Forget class, and 1024 samples for the Remain class, from the training sets of CIFAR10 and CIFAR100's, for computing $r_{l,j}^c$ for each filter. For VGG-16, we only consider the last 8 layers, and for ResNet56, we only consider the final layerblock. We then select the sparsity budgets $B_l$, and prune the most discriminative filters for that layer. We also fine-tune the models for 1 epoch. We measure the accuracy on the class test set, and the class complement test sets both before and after fine-tuning, and we compare against models trained from scratch on the retain set (the class complement).

**Results:** We present a summary of our results in Table 1. In particular, we highlight that on CIFAR10 models, particularly VGG16, the performance is comparable to or outperforms baselines with minimal fine-tuning. There are two interesting observations: first, as the number of classes exceeds the width of the network (as was the case with ResNet56 trained on CIFAR100), the efficacy of our method is drastically affected. Second, fine-tuning models on the remain set still raises the accuracy on the forget set unless dramatically more filters are pruned. The reasons for this will be the focus of future work.

This set of experiments highlights the fact that for classifier models, it is possible to identify subnetworks responsible for predictions for each class. Identifying these subnetworks then facilitates classwise unlearning.

### 6.2 Structured Pruning with DISCEDIT-SP with Fine-Tuning

In this section, we investigate the ability for DISCEDIT-SP to sparsify models effectively both with and without fine-tuning.

**Experiment Setup:** We prune VGG16, VGG19, and ResNet56 models trained on CIFAR10 with two sets of fixed sparsity budgets, and then fine-tune them for 50 epochs. We repeat the experiment for a ResNet50 trained on Imagenet, and fine-tune the pruned models for 100 epochs. We choose $f(x) = u^\top \varphi(x)$, where $\varphi(x) = [1^\top x, (1^\top x)^2]^\top$ as our witness function in each of the experiments. For details about the pretrained models used, refer to Appendix F.

**Results:** We show that models pruned with DISCEDIT-SP without fine-tuning retain high accuracies, particularly on the CIFAR10 dataset. Moreover, after fine-tuning, DISCEDIT-SPis

**Table 2:** DISCEDIT-SP performance on CIFAR10 and ImageNet models with fine-tuning. TVSPrune refers to [39], and CHIP refers to [52]. 'Sparsity' refers to parametric sparsity.

| CIFAR-10 | | | | | |
|---|---|---|---|---|---|
| Model | Sparsity | our work | TVSPrune | CHIP | L1 |
| VGG16 | 61.2% | **-0.37%** | -0.98% | -0.73% | 1.26% |
| | 75.05% | **-1.32%** | -1.54% | -1.62 | -2.31 |
| VGG19 | 72.4% | **-0.12%** | -0.16% | N/A | -2.41% |
| | 76.1% | **-0.96%** | -1.13% | N/A | -3.30% |
| ResNet56 | 60.7% | **-1.21%** | -1.92% | -1.77% | -6.21% |
| ImageNet | | | | | |
| Model | Sparsity | our work | TVSPrune | CHIP | L1 |
| ResNet50 | 20.2% | **+0.12%** | -0.4% | **+0.10%** | -1.08% |
| | 40.8% | **-2.58%** | -2.74% | -2.76% | -4.45% |

able to almost fully recover the accuracy of the original models. Our results are summarized in Table 2, which shows the drop in accuracy achieved by the different pruning algorithms compared after fine-tuning.

## 7 Conclusions

In this work, we address the problem of model editing by analyzing discriminative properties of the feature maps. We leverage the notion of discriminative components to derive algorithms for two relevant tasks: structured pruning and class unlearning. Additionally, in order to identify discriminative components, we derive new lower bounds on the TV distance. These lower bounds also elucidate previously unknown connections between the Total Variation distance and discriminative classifiers, specifically the Fisher discriminant and the Minimax Probability Machine. Our experiments show that the model editing algorithms derived by our methods are highly effective, and match or outperform current state of the art in structured pruning, and can reduce accuracy almost completely on a given class, while maintaining accuracy on the remaining classes. This work, however, currently analyzes discriminative components (and thus, subnetworks responsible for classwise predictions) in classifier models. Current avenues of research include extending these results to unlearning and pruning of generative models as well. Lastly, the techniques proposed in this work can be extended to other editing tasks as well, such as debiasing.

## Acknowledgments

The authors gratefully thank **Shell India Markets Pvt Ltd**, for their generous support and contributions to this work.

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

# Appendix

In this appendix, we provide the following material.

- In Appendix A, we provide a brief survey of the literature that addresses both structured pruning.

- In Appendix B, we provide proofs for the theoretical results proposed in this work.

- In Appendix C, we provide additional analysis of the practical performance of our algorithm.

- In Appendix D, we provide details for variants of the DISCEDIT-SP algorithm, such as the DISCEDIT-SP-F, DISCEDIT-SP-M, DISCEDIT-SP-R, and DISCEDIT-SP-E variants. We also discuss the use of the BatchNorm random variables as opposed to the entire feature map, in order to reduce memory storage, as well as deriving TVSPrune [39] using our results.

- In Appendix E, we provide additional experiments, including more detailed results from experiments conducted in Section 6.

- In Appendix F, we provide additional details, such as hyperparameters and dataset splits, used in this work.

## A  Related Work

### A.1  Structured Pruning

In this section, we detail related work in the space of structured pruning Pruning is a long-standing strategy for reducing the inference time and power consumption of neural networks [29, 18]. Pruning algorithms can be divided into structured - wherein entire filters or neurons are removed - or unstructured - wherein individual weights are removed - techniques [19, 4]. We refer readers to [19, 4, 11, 12], and the references therein, for a detailed discussion on unstructured pruning methods.

#### A.1.1  Structured Pruning that Requires the Loss Function

Structured pruning methods are of interest as they enable the reduction of inference times without requiring specialized hardware or software [19, 38, 41]. Pruning algorithms require saliency measures that quantify the importance of filters or neurons, and prune the unimportant ones [19]. Pruning algorithms where access to the loss function is not only assumed, but is necessary, generate saliencies using gradient information [38, 37, 43, 50], and the loss function Hessians, or approximations thereof [29, 18, 36, 60, 55, 54]. For more similar works, we refer readers to the surveys [19, 4, 6].

#### A.1.2  Structured Pruning without the Loss Function

In this section, we discuss structured pruning methods that do not require derivatives of the loss function, aside from fine-tuning. There are a variety of approaches to solve the problem of identifying which filters to prune without using the loss function. Works such as [31, 32] use the norms of the weights to identify filters to prune. Other works, such as [41, 59], identify which filters to prune by bounding the reconstruction error incurred by pruning a given filter. Works such as [35, 52] analyze the rank of feature maps to measure filter importance. Another body of work uses coresets of feature maps to identify subsets of important filters, as described in works such as [34, 53, 40]. Recently, some works propose identifying important filters by their discriminative ability. Recently, in [20, 21], discrimination-aware metrics were used to identify filters important to classification accuracy. In [39], the class conditional distributions were assumed to be Gaussian, and the Hellinger distance between the class conditional feature maps was used to identify important filters. In this work, we formalize the notion of discriminative ability in terms of the Bayes risk, and derive novel lower bounds on the TV distance to effectively approximate it. Furthermore, we emphasize that this work is clearly differentiated from prior art in this area in these critical respects: first, no assumptions are required of the class conditional distributions; second, the discriminative ability of a filter is formalized in terms of the Bayes error of a classifier trained on the feature maps it yields; third, the pruning algorithms we derive using our method require no access to the training data or loss function.

## A.2 Machine Unlearning

In this section, we provide a detailed literature survey on machine unlearning, both with and without model editing.

### A.2.1 Machine Unlearning without Model Editing

Machine unlearning has gained importance in recent years owing to data privacy and security concerns [5, 42]. A wide variety of works exist to address this problem. Several works aim to forget data points, even in the adaptive setting, while maintaining the accuracy of the model, such as [47, 16, 22, 14]. The work in [47] also provides bounds on the number of samples that a model can be allowed to forget before accuracy degradation. Machine unlearning is also a significant area of research in the space of large language models, as noted in [26, 10], and generative models [13].

Another aspect of machine unlearning is selective forgetting, wherein classes, groups, or sets of samples are forgotten from the network, as described in [58] and the references therein. This connects machine unlearning to the continual learning setting as well, as described in [57] and the references cited there.

### A.2.2 Machine Unlearning with Model Editing

While there has been significant research into selective forgetting and model unlearning, facilitating selective forgetting and machine unlearning by *editing* models remains an underexplored field. Recent works such as [23, 46] explore the effect of sparsity on machine unlearning and continual learning; in particular, [23] shows that sparsifying models prior to unlearning can increase the effectiveness of it. However, recent work such as [56] investigates using pruning for model unlearning in the federated setting. More recently, [48] uses pruning to selectively forget a single class from a ResNet50 imagenet model with minimal loss in accuracy. Our work differs from prior work in this space since we directly use the discriminative ability of model components to identify which components to remove to forget a given class.

# B  Proofs of Main Results

In this section, we provide the proofs for the main theoretical results proposed in the paper. Specifically, we provide proofs for Theorems 1-3 and Corollary 1.

## B.1  Proof of Theorem 1

In this section, we provide the proofs for Theorem 1.

**Theorem.** *Let $\mathbb{P}, \mathbb{Q}$ be two probability measures supported on $X \subseteq \mathbb{R}^d$, and let $p$ and $q$ be the corresponding densities. Let $\mathcal{F}$ be the set of functions with bounded first and second moments defined on $X$. Then,*

$$\mathrm{TV}(\mathbb{P}, \mathbb{Q}) \geq \sup_{f \in \mathcal{F}} \frac{\left(\bar{f}_p - \bar{f}_q\right)^2}{2\left(\bar{f}_p^{(2)} + \bar{f}_q^{(2)}\right)} \tag{11}$$

*Proof.* Choose an arbitrary $f \in \mathcal{F}$. Then, we have

$$
\begin{aligned}
\left(\bar{f}_p - \bar{f}_q\right)^2 &= \left(\int_{\mathcal{X}} (p(x) - q(x)) f(x) dx\right)^2 \\
&= \left(\int_{\mathcal{X}} \left(\sqrt{|p(x) - q(x)|} \sqrt{|p(x) - q(x)|}\right) f(x) dx\right)^2 \\
&\leq \left(\sqrt{\int_{\mathcal{X}} |p(x) - q(x)| dx}\right)^2 \left(\sqrt{\int_{\mathcal{X}} |p(x) - q(x)| f(x)^2 dx}\right)^2 \quad \text{(by Cauchy-Schwarz)} \\
&= 2\mathrm{TV}(\mathbb{P}, \mathbb{Q}) \left(\int_{\mathcal{X}} |p(x) - q(x)| f(x)^2 dx\right) \quad \text{(by Definition 1)} \\
&\leq 2\mathrm{TV}(\mathbb{P}, \mathbb{Q}) \left(\int_{\mathcal{X}} (p(x) + q(x)) f(x)^2 dx\right) \\
&= 2\mathrm{TV}(\mathbb{P}, \mathbb{Q}) \left(\bar{f}_p^{(2)} + \bar{f}_q^{(2)}\right).
\end{aligned}
$$

Thus, for any arbitrary $f \in \mathcal{F}$, we have

$$
2\mathrm{TV}(\mathbb{P}, \mathbb{Q}) \geq \frac{\left(\bar{f}_p - \bar{f}_q\right)^2}{\left(\bar{f}_p^{(2)} + \bar{f}_q^{(2)}\right)},
$$

from which it follows that

$$
2\mathrm{TV}(\mathbb{P}, \mathbb{Q}) \geq \sup_{f \in \mathcal{F}} \frac{\left(\bar{f}_p - \bar{f}_q\right)^2}{\left(\bar{f}_p^{(2)} + \bar{f}_q^{(2)}\right)}. \tag{12}
$$

$\square$

The proof of Theorem 1 follows from the fact that we can choose $f(x) = u^\top \varphi(x)$, and apply the formula for the expectation. Then, maximizing over $\mathcal{F}$ is equivalent to maximizing over $u$. Thus, the proof is completed.

## B.2 Proof of Theorem 2

First, we restate the Theorem for clarity.

**Theorem.** *Suppose $\mathbf{P}$ and $\mathbf{Q}$ are sets of moments of two probability measures supported on $\mathcal{X}$. Let $\varphi(X)$ be a vector of polynomials such that $\mathbb{E}_{\mathbb{P}}[\varphi(X)] = g(\mathbf{P})$, $\mathbb{E}_{\mathbb{Q}}[\varphi(X)] = g(\mathbf{Q})$, $\mathbb{E}_{\mathbb{P}}[\varphi(X)\varphi(X)^\top] = G(\mathbf{P})$, and $\mathbb{E}_{\mathbb{Q}}[\varphi(X)\varphi(X)^\top] = G(\mathbf{Q})$, and let $f = u^\top (\varphi(X) - \frac{g(\mathbf{P}) - g(\mathbf{Q})}{2})$, be a witness function. Then, for any $\mathbb{P} \in \mathcal{S}_k(\mathbf{P})$, $\mathbb{Q} \in \mathcal{S}_k(\mathbf{Q})$, supported on a set $\mathcal{X} \subseteq \mathbb{R}^d$, we have*

$$
D_{\mathrm{TV}}(\mathbf{P}, \mathbf{Q}; k) \geq \frac{S_{\mathrm{TV}}^*(\mathbf{P}, \mathbf{Q})}{2 + S_{\mathrm{TV}}^*(\mathbf{P}, \mathbf{Q})} \tag{13}
$$

*where*

$$
S_{\mathrm{TV}}^*(\mathbf{P}, \mathbf{Q}) = (\Delta g)^\top (\tilde{G}(\mathbf{P}) + \tilde{G}(\mathbf{Q}))^{-1} (\Delta g) \tag{14}
$$

*and $\Delta g = g(\mathbf{P}) - g(\mathbf{Q})$ and $\tilde{G}(\mathbf{P}) = G(\mathbf{P}) - g(\mathbf{P})g(\mathbf{P})^\top$.*

*Proof.* First, recall that Theorem 1 provides lower bounds for *all* distributions for which the moments of the witness function are given by $\bar{f}_p$, $\bar{f}_p^{(2)}$, $\bar{f}_q$, $\bar{f}_q^{(2)}$. Begin by choosing

$$
f = u^\top \left(\varphi(x) - \frac{g(\mathbf{P}) + g(\mathbf{Q})}{2}\right),
$$

where $u \in \mathbb{R}^d$ is constant. Then,

$$
\bar{f}_p = \frac{1}{2} u^\top (g(\mathbf{P}) - g(\mathbf{Q})) \quad \text{and} \quad \bar{f}_q = \frac{1}{2} u^\top (g(\mathbf{Q}) - g(\mathbf{P})),
$$

and

$$\bar{f}_p^{(2)} = u^\top \left( \mathbb{E}_\mathbb{P} \left[ \left( \varphi(x) - \frac{g(\mathbf{P}) + g(\mathbf{Q})}{2} \right) \left( \varphi(x) - \frac{g(\mathbf{P}) + g(\mathbf{Q})}{2} \right)^\top \right] \right) u$$

$$= u^\top \left( \mathbb{E}_\mathbb{P} \left[ (\varphi(x) - \Delta) (\varphi(x) - \Delta)^\top \right] \right) u \quad \left( \text{setting } \Delta = \frac{g(\mathbf{P}) + g(\mathbf{Q})}{2} \right)$$

$$= u^\top \left( \mathbb{E}_\mathbb{P} \left[ (\varphi(x) - g(\mathbf{P}) + g(\mathbf{P}) - \Delta) (\varphi(x) - g(\mathbf{P}) + g(\mathbf{P}) - \Delta)^\top \right] \right) u$$

$$= u^\top \left( G(\mathbf{P}) - g(\mathbf{P})g(\mathbf{P})^\top + \frac{1}{4}(g(\mathbf{P}) - g(\mathbf{Q}))(g(\mathbf{P}) - g(\mathbf{Q}))^\top \right) u.$$

Similarly, we have

$$\bar{f}_q^{(2)} = u^\top \left( G(\mathbf{Q}) - g(\mathbf{Q})g(\mathbf{Q})^\top + \frac{1}{4}(g(\mathbf{P}) - g(\mathbf{Q}))(g(\mathbf{P}) - g(\mathbf{Q}))^\top \right) u.$$

Substituting this into (11), we get

$$\mathrm{TV}(\mathbb{P}, \mathbb{Q}) \geq \sup_{u \in \mathbb{R}^d} \frac{1}{2} \frac{\left( u^\top (g(\mathbf{P}) - g(\mathbf{Q})) \right)^2}{u^\top (G(\mathbf{P}) - g(\mathbf{P})g(\mathbf{P})^\top + G(\mathbf{Q}) - g(\mathbf{Q})g(\mathbf{Q})^\top) u + \frac{1}{2} \left( u^\top (g(\mathbf{P}) - g(\mathbf{Q})) \right)^2}$$

$$= \sup_{u \in \mathbb{R}^d} \frac{1}{2} \frac{2 \left( u^\top (g(\mathbf{P}) - g(\mathbf{Q})) \right)^2}{2u^\top (G(\mathbf{P}) - g(\mathbf{P})g(\mathbf{P})^\top + G(\mathbf{Q}) - g(\mathbf{Q})g(\mathbf{Q})^\top) u + \left( u^\top (g(\mathbf{P}) - g(\mathbf{Q})) \right)^2}$$

$$= \sup_{u \in \mathbb{R}^d} \frac{\frac{\left( u^\top (g(\mathbf{P}) - g(\mathbf{Q})) \right)^2}{u^\top (\tilde{G}(\mathbf{P}) + \tilde{G}(\mathbf{Q})) u}}{2 + \frac{\left( u^\top (g(\mathbf{P}) - g(\mathbf{Q})) \right)^2}{u^\top (\tilde{G}(\mathbf{P}) + \tilde{G}(\mathbf{Q})) u}}$$

$$= \sup_{u \in \mathbb{R}^d} \frac{\frac{\left( u^\top (g(\mathbf{P}) - g(\mathbf{Q})) \right)^2}{u^\top (\tilde{G}(\mathbf{P}) + \tilde{G}(\mathbf{Q})) u}}{2 + \frac{\left( u^\top (g(\mathbf{P}) - g(\mathbf{Q})) \right)^2}{u^\top (\tilde{G}(\mathbf{P}) + \tilde{G}(\mathbf{Q})) u}}$$

Let

$$S_{\mathrm{TV}}(\mathbf{P}, \mathbf{Q}; u) = \frac{\left( u^\top (g(\mathbf{P}) - g(\mathbf{Q})) \right)^2}{u^\top (\tilde{G}(\mathbf{P}) + \tilde{G}(\mathbf{Q})) u}.$$

Then,

$$S_{\mathrm{TV}}^* = \max_u S_{\mathrm{TV}}(\mathbf{P}, \mathbf{Q}; u) = \frac{\left( u^\top (g(\mathbf{P}) - g(\mathbf{Q})) \right)^2}{u^\top (\tilde{G}(\mathbf{P}) + \tilde{G}(\mathbf{Q})) u} = (\Delta g)^\top (\tilde{G}(\mathbf{P}) + \tilde{G}(\mathbf{Q}))^{-1} (\Delta g)$$

Thus, we have

$$\mathrm{TV}(\mathbb{P}, \mathbb{Q}) \geq \frac{S^*(\mathbf{P}, \mathbf{Q})}{2 + S^*(\mathbf{P}, \mathbf{Q})} \tag{15}$$

To show the second inequality, we first state the following Lemma.

**Lemma 1.** *Suppose $\mathbf{P}$ and $\mathbf{Q}$ are sets of moments of two probability measures supported on $\mathcal{X}$. Let $\varphi(X)$ be a vector of polynomials such that $\mathbb{E}_\mathbb{P}[\varphi(X)] = g(\mathbf{P})$, $\mathbb{E}_\mathbb{Q}[\varphi(X)] = g(\mathbf{Q})$, $\mathbb{E}_\mathbb{P}[\varphi(X)\varphi(X)^\top] = G(\mathbf{P})$, and $\mathbb{E}_\mathbb{Q}[\varphi(X)\varphi(X)^\top] = G(\mathbf{Q})$, and let*

$$S_{\mathrm{H}}(\mathbf{P}, \mathbf{Q}; u) = \frac{|u^\top (g(\mathbf{P}) - g(\mathbf{Q}))|}{\sqrt{u^\top G(\mathbf{P}) u} + \sqrt{u^\top G(\mathbf{Q}) u}} \text{ and } S_{\mathrm{TV}}(\mathbf{P}, \mathbf{Q}; u) = \frac{\left( u^\top (g(\mathbf{P}) - g(\mathbf{Q})) \right)^2}{u^\top (G(\mathbf{P}) u + G(\mathbf{Q})) u},$$

*and let*

$$S_{\mathrm{H}}^\star(\mathbf{P}, \mathbf{Q}) = \arg\max_u S_{\mathrm{H}}(\mathbf{P}, \mathbf{Q}; u) \text{ and } S_{\mathrm{TV}}^\star(\mathbf{P}, \mathbf{Q}) = \arg\max_u S_{\mathrm{TV}}(\mathbf{P}, \mathbf{Q}; u).$$

*Then,*

$$\left( \frac{S_{\mathrm{H}}^*(\mathbf{P}, \mathbf{Q})}{\sqrt{2} + S_{\mathrm{H}}^*(\mathbf{P}, \mathbf{Q})} \right)^2 \leq \frac{S_{\mathrm{TV}}^*(\mathbf{P}, \mathbf{Q})}{2 + S_{\mathrm{TV}}^*(\mathbf{P}, \mathbf{Q})}. \tag{16}$$

*Proof.* To prove this statement, we first show that $S_{\mathrm{H}}^*(\mathbf{P},\mathbf{Q})^2 \leq S_{\mathrm{TV}}^*(\mathbf{P},\mathbf{Q})$. Fix

$$\hat{u} = \arg\max_u S_{\mathrm{H}}^*(\mathbf{P},\mathbf{Q};u).$$

From this, we see that

$$S_{\mathrm{H}}^*(\mathbf{P},\mathbf{Q})^2 = \frac{(\hat{u}^\top(g(\mathbf{P})-g(\mathbf{Q})))^2}{\left(\sqrt{\hat{u}^\top G(\mathbf{P})\hat{u}} + \sqrt{\hat{u}^\top G(\mathbf{Q})\hat{u}}\right)^2} = \frac{(\hat{u}^\top(g(\mathbf{P})-g(\mathbf{Q})))^2}{\hat{u}^\top G(\mathbf{P})\hat{u} + \hat{u}^\top G(\mathbf{Q})\hat{u} + 2\sqrt{\hat{u}^\top G(\mathbf{P})\hat{u}}\sqrt{\hat{u}^\top G(\mathbf{Q})\hat{u}}}$$

$$\leq \frac{\left(\hat{u}^\top(g(\mathbf{P})-g(\mathbf{Q}))\right)^2}{\hat{u}^\top\left(G(\mathbf{P})u + G(\mathbf{Q})\right)\hat{u}} \leq S_{\mathrm{TV}}^*(\mathbf{P},\mathbf{Q}).$$

Next, let

$$d_1 = \frac{S_{\mathrm{TV}}^*(\mathbf{P},\mathbf{Q})}{2 + S_{\mathrm{TV}}^*(\mathbf{P},\mathbf{Q})} \text{ and } 2d_2 = \left(\frac{S_{\mathrm{H}}^*(\mathbf{P},\mathbf{Q})}{\sqrt{2} + S_{\mathrm{H}}^*(\mathbf{P},\mathbf{Q})}\right)^2.$$

We have

$$d_2 = \frac{S_{\mathrm{H}}^*(\mathbf{P},\mathbf{Q})^2}{2 + S_{\mathrm{H}}^*(\mathbf{P},\mathbf{Q})^2 + 2\sqrt{2}S_{\mathrm{H}}^*(\mathbf{P},\mathbf{Q})} \leq \frac{S_{\mathrm{H}}^*(\mathbf{P},\mathbf{Q})^2}{2 + S_{\mathrm{H}}^*(\mathbf{P},\mathbf{Q})^2}$$

Since $\frac{x}{x+2}$ is monotonically increasing for positive $x$, we prove the statement by the fact that $S_{\mathrm{TV}}^* \geq S_{\mathrm{H}}^{*\,2}$. $\square$

With these results, the Theorem is proved. $\square$

## B.3 Proof of Corollary 1: Computing the Bayes Classifier and $\mathrm{TV}(\mathbb{P},\mathbb{Q})$ from the Lower Bound

In this section, we prove Corollary 1. Recall that the lower bound proposed in Theorem 1 is not tight, as the Cauchy-Schwarz inequality used in the derivation of the bound is only not strict when the witness function $f$ is a constant. However, there are cases where the bound can be used to compute the true TV distance. We show this case in Corollary 1, which we restate and prove in the sequel.

**Corollary.** *Suppose* $\mathbb{P} \equiv \mathcal{N}(\mu_p, \Sigma)$ *and* $\mathbb{Q} \equiv \mathcal{N}(\mu_q, \Sigma)$ *Let* $f(x;u) = u^\top(x - \frac{1}{2}(\mu_p - \mu_q))$ *be a witness function. Then,*

$$\mathrm{TV}(\mathbb{P},\mathbb{Q}) = 2\Phi\left(\sqrt{(u^*)^\top(\mu_p - \mu_q)/2}\right) - 1,$$

*where*

$$u^* = \arg\max_u \frac{(\mathbb{E}_{x\sim\mathbb{P}}[f(x;u)] - \mathbb{E}_{x\sim\mathbb{Q}}[f(x;u)])^2}{\mathbb{E}_{x\sim\mathbb{P}}[f(x;u)^2] + \mathbb{E}_{x\sim\mathbb{Q}}[f(x;u)^2]}$$

*Proof.* First, following the proof of Corollary 2, we have $u^* = \Sigma^{-1}(\mu_p - \mu_q)$. Note that this is identical to the weights of the Gaussian discriminant classifier discussed in, say, [9]. Substituting $u*$ into the expression $\mathrm{TV}(\mathbb{P},\mathbb{Q}) = 2\Phi\left(\sqrt{(u^*)^\top(\mu_p - \mu_q)/2}\right) - 1$, we get $\mathrm{TV}(\mathbb{P},\mathbb{Q}) = 2\Phi\left(\sqrt{(\mu_p - \mu_q)^\top\Sigma^{-1}(\mu_p - \mu_q)/2}\right) - 1$. The risk of the Bayes classifier, as given in [45], is $R^*(\mathbb{P},\mathbb{Q}) = \Phi(-\sqrt{(\mu_p - \mu_q)^\top\Sigma^{-1}(\mu_p - \mu_q)}/2)$, where $\Phi(x)$ is the Gaussian cdf. Using the fact that $\Phi(x) = 1 - \Phi(-x)$, and the identity $2R^*(\mathbb{P},\mathbb{Q}) = 1 - \mathrm{TV}(\mathbb{P},\mathbb{Q})$, we get $\mathrm{TV}(\mathbb{P},\mathbb{Q}) = 2\Phi\left(\sqrt{(\mu_p - \mu_q)^\top\Sigma^{-1}(\mu_p - \mu_q)/2}\right) - 1$. Note that with this choice of $u*$, the square root term remains well-defined. This matches the well-known result for the TV distance between Gaussian measures with the same variance. Thus, we prove the statement. $\square$

*Remark:* This result also illustrates the case where the Bayes' classifier lies in the set of functions $\mathcal{F} := \{f(x) : f(x) = u^\top\varphi(x)\}$ for a given function $\varphi(x)$. In this case, if $\varphi(x) = x - \frac{1}{2}(\mu_p - \mu_q)$, and $\mathbb{P}$ and $\mathbb{Q}$ are Gaussian with the same variant, the Bayes classifier is equivalent to the Fisher discriminant.

### B.4 Proof of Theorem 3

In this section, we prove Theorem 3. We restate the result for convenience, and prove the Theorem thereafter.

**Theorem.** *Suppose* $\mathbb{P}, \mathbb{Q}$ *be two probability measures supported on* $X \subseteq \mathbb{R}^d$, *with densities* $p$ *and* $q$, *and let* $\mu_p = \mathbb{E}_{\mathbb{P}}[x]$, $\mu_q = \mathbb{E}_{\mathbb{Q}}[x]$ *and* $C_p = \mathbb{E}_{\mathbb{P}}[xx^\top]$, $C_q = \mathbb{E}_{\mathbb{Q}}[xx^\top]$. *Suppose we have plug-in estimates* $\bar{\mu}_p, \bar{C}_p, \bar{\mu}_q, \bar{C}_q$ *as defined in* (7), *that satisfy*

$$\|\mu_p - \bar{\mu}_p\|_2 \le \delta_p \text{ and } \|\mu_q - \bar{\mu}_q\|_2 \le \delta_q$$
$$\|C_p - \bar{C}_p\|_F \le \rho_p \text{ and } \|C_q - \bar{C}_q\|_F \le \rho_q.$$

*Then, with a witness function of the form* $f(x) = u^\top x$

$$D_{\text{TV}}(\mathbb{P}, \mathbb{Q}) \ge \min_{\mu_p, \mu_q \in \mathcal{M}} (\Delta\mu)^\top (C_p + C_q + \rho I)^{-1}(\Delta\mu), \tag{17}$$

*where* $\mathcal{M} = \{(\mu_p, \mu_q) : \|\mu_p - \bar{\mu}_p\|_2 \le \delta_p, \|\mu_q - \bar{\mu}_q\|_2 \le \delta_q\}$, $\Delta\mu = \mu_p - \mu_q$, *and* $\rho = \rho_p + \rho_q$.

*Proof.* First, note that $f(x) = u^\top x$. Thus,

$$\bar{f}_p = u^\top \mu_p \text{ and } \bar{f}_q = u^\top \mu_q$$
$$\bar{f}_p^{(2)} = u^\top C_p u \text{ and } \bar{f}_{(q)} = u^\top C_q u.$$

Thus, for exact values, we have

$$\text{TV}(\mathbb{P}, \mathbb{Q}) \ge D_{\text{TV}}(\mathbb{P}, \mathbb{Q}; 2) = \frac{(u^\top(\mu_p - \mu_q))^2}{u^\top(C_p + C_q)u} = (\mu_p - \mu_q)^\top (C_p + C_q)^{-1}(\mu_p - \mu_q).$$

However, there are errors that arise from using plug-in estimators (defined in (7). To handle the estimation error for $C_p$ and $C_q$, we rely on (23) in Lanckriet et al. [28]. That is,

$$\max_{C_p : \|\bar{C}_p - C_p\|_F \le \rho} u^\top C_p u = u^\top (\bar{C}_p + \rho I)u.$$

We substitute these values back into the lower bound, and impose the constraints on the $\mu_p$ and $\mu_q$. Thus, we prove the theorem. ☐

*Remark* B.1. The proof follows a similar logic to that of the derivation of (15) in Kim et al. [24].

## C  Additional Results: Computational Complexity of $r_{l,j}$ scores with Different Witness functions

In this section, we detail the computational cost of computing the $r_{l,j}$ scores needed for DISCEDIT-SP and DISCEDIT-U. We tabulate our results in Table 3.

**Explanation for Storage Complexity**   For a fixed witness function, for each filter, we only need to store 4 real numbers per class (that is, for a distribution $P$, we store $\bar{f}_c$, $\bar{f}_c^{(2)}$, $\bar{f}_{\bar{c}}$, and $\bar{f}_{\bar{c}}^{(2)}$. For Fisher and MPM type witness functions, we need to store the class-conditional and class-complement means and covariances of $\varphi(X)$ for each class, in total requiring the storage of $O(n^2)$ values. For DISCEDIT-U, since we only need to compute scores for a single class, the dependence on $C$ vanishes.

**Explanation for Computation Complexity**   The computational complexity of one-shot pruning of a given layer using DISCEDIT-SPdepends on the number of filters $L$, the number of classes in the dataset $C$, and the witness function itself. If the Witness function is fixed a priori, the cost of computing each TV distance is $O(1)$. Thus, in this case, the complexity is $O(LC)$ to compute all the pairwise TV distance lower bounds, and $O(LC)$ to find the minimum for each filter. For Fisher discriminant-type scores (such as are used in DISCEDIT-SP-F, or the witness function used in Section 7.3), given that $\varphi(X)$ is a vector of length $n$, this requires the inversion of a matrix, which requires $O(n^2)$ iterations. Thus, in this case, the cost is $O(LCn^2)$ iterations (this subsumes the cost of finding the minimum). For MPM based witness functions (such as those used in DISCEDIT-SP-Q), this requires solving an SOCP ($\tilde{O}(n^3)$ complexity, $\tilde{O}(\cdot)$ suppresses $\epsilon$ accuracy terms). Thus, the cost is $\tilde{O}(LCn^3)$. For DISCEDIT-U, since we only need to compute scores for a single class, the dependence on $C$ vanishes.

**Table 3:** Comparison of complexities of computing different witness functions. (P) refers to pruning (used in DISCEDIT-SP) and (U) refers to unlearning (used in DISCEDIT-U).

| Witness Function | Cost (P) | Storage (P) | Cost (U) | Storage (U) |
|---|---|---|---|---|
| **Fixed (a priori)** | $O(LC)$ | $O(LC)$ | $O(L)$ | $O(L)$ |
| **Fisher-type Witness Function** | $O(LCn^2)$ | $O(LCn^2)$ | $O(Ln^2)$ | $O(Ln^2)$ |
| **MPM-type Witness Function** | $O(LCn^3)$ | $O(LCn^2)$ | $O(Ln^3)$ | $O(Ln^2)$ |

# D  Variants of the DISCEDIT-SP Algorithm

In this section, we propose a variety of variants of DISCEDIT-SPalgorithm. In particular, we highlight how changing the witness function used to lower bound the total variation distance can lead to new algorithms. Moreover, we show how TVSPrune [39] can be recovered from DISCEDIT-SP; indeed, it is a special case of it.

## D.1  Fisher-based Lower Bounds and TVSPrune

We choose $f(X) = u^\top \varphi(X)$. Let

$$\mu_{j,c}^l = \mathbb{E}_{X \sim \mathcal{D}_c} \left[ \varphi(X) \right]$$

and let

$$\Sigma_{j,c}^l = \mathbb{E}_{X \sim \mathcal{D}_c} \left[ (\varphi(X) - \mu_{j,c}^l)(\varphi(X) - \mu_{j,c}^l)^\top \right].$$

Then, we get

---

**Algorithm 2:** DISCEDIT-SP-F

---

**Input:** Class conditional distributions $\mathcal{D}_c$, $c \in [C]$, pretrained CNN with parameters
    $\mathcal{W} = (W_1, \cdots, W_L)$, layerwise sparsity budgets $B^l$, witness function $f$
**for** $l \in [L]$ **do**
    Set $S^l = [s_1^l, \cdots, s_{N_l}^l] = \mathbf{0}_{N_l}$
    Compute $\mu_{j,c}^l \, \mu_{j,\bar{c}}^l, \Sigma_{j,c}^l, \Sigma_{j,\bar{c}}^l$ for all $j, c$
    Compute $r_j^l = \min_c \mathsf{Fish}^\star(\mathcal{D}_{j,c}^l, \mathcal{D}_{j,\bar{c}}^l)$ for all $j$.
    **if** $j \in \mathrm{sort}_{B_l}(\{r_j^l\}_{j=1}^{N_l})$ **then**
        Set $s_j^l = 1$
**Output:** Sparse Masks $S_1, \cdots, S^L$
**return** $\hat{\mathcal{W}}$

---

If $\varphi(X)$ is a vector of quadratic functions of $X$, we call the algorithm  DISCEDIT-SP-FQ.

## D.2  Minimax Probability Machine based Algorithms

We define $\mu_{j,c}^l \, \mu_{j,\bar{c}}^l, \Sigma_{j,c}^l, \Sigma_{j,\bar{c}}^l$ as previously. We then state the algorithm as follows.

---

**Algorithm 3:** DISCEDIT-SP-M

---

**Input:** Class conditional distributions $\mathcal{D}_c$, $c \in [C]$, Pretrained CNN with parameters
    $\mathcal{W} = (W_1, \cdots, W_L)$, layerwise sparsity budgets $B^l$, witness function $f$
**for** $l \in [L]$ **do**
    Set $S^l = [s_1^l, \cdots, s_{N_l}^l] = \mathbf{0}_{N_l}$
    Compute $\mu_{j,c}^l \, \mu_{j,\bar{c}}^l, \Sigma_{j,c}^l, \Sigma_{j,\bar{c}}^l$ for all $j, c$
    Compute $r_j^l = \min_c \mathsf{MPM}^\star(\mathcal{D}_{j,c}^l, \mathcal{D}_{j,\bar{c}}^l)$ for all $j$.
    for all $j$.
    **if** $j \in \mathrm{sort}_{B_l}(\{r_j^l\}_{j=1}^{N_l})$ **then**
        Set $s_j^l = 1$
**Output:** Sparse Masks $S_1, \cdots, S^L$
**return** $\hat{\mathcal{W}}$

---

If $\varphi(X)$ is a vector of quadratic functions of $X$, we call the algorithm  DISCEDIT-SP-MQ.

## D.3 EnsemblePrune - Taking the best of FisherPrune and MPMPrune

We choose the same witness functions as we did for DISCEDIT-SP-M and DISCEDIT-SP-F, and define the moments in the same fashion. We then

---

**Algorithm 4:** DISCEDIT-SP-E

---

**Input:** Class conditional distributions $\mathcal{D}_c$, $c \in [C]$, Pretrained CNN with parameters
$\qquad \mathcal{W} = (W_1, \cdots, W_L)$, layerwise sparsity budgets $B^l$, witness function $f$

**for** $l \in [L]$ **do**

$\qquad$ Set $S^l = [s_1^l, \cdots, s_{N_l}^l] = \mathbf{0}_{N_l}$

$\qquad$ Compute $\mu_{j,c}^l$ $\mu_{j,\bar{c}}^l$, $\Sigma_{j,c}^l$, $\Sigma_{j,\bar{c}}^l$ for all $j, c$

$\qquad$ Compute

$$r_j^l = \max \left\{ \max_u \frac{|u^\top(\mu_{j,c}^l - \mu_{j,\bar{c}}^l)|}{\sqrt{u^\top \Sigma_{j,c}^l u} + \sqrt{u^\top \Sigma_{j,\bar{c}}^l u}}, \ \max_u \frac{(u^\top(\mu_{j,c}^l - \mu_{j,\bar{c}}^l)^2)}{2u^\top(\Sigma_{j,c}^l + \Sigma_{j,\bar{c}}^l)^2)u} \right\}$$

$\qquad$ for all $j$.

$\qquad$ **if** $j \in \mathrm{sort}_{B_l}(\{r_j^l\}_{j=1}^{N_l})$ **then**

$\qquad\qquad$ Set $s_j^l = 1$

**Output:** Sparse Masks $S_1, \cdots, S^L$

**return** $\hat{\mathcal{W}}$

---

## D.4 RobustPrune - Accounting for Errors in Moment Measurements

We choose the same witness functions as we did for DISCEDIT-SP-M and DISCEDIT-SP-F, and define the moments in the same fashion. We then derive the following algorithm.

---

**Algorithm 5:** DISCEDIT-SP-R

---

**Input:** Class conditional distributions $\mathcal{D}_c$, $c \in [C]$, pretrained CNN with parameters
$\qquad \mathcal{W} = (W_1, \cdots, W_L)$, layerwise sparsity budgets $B^l$, witness function $f = u^\top \varphi(x)$, error
$\qquad$ tolerance $\gamma$

**for** $l \in [L]$ **do**

$\qquad$ Set $S^l = [s_1^l, \cdots, s_{N_l}^l] = \mathbf{0}_{N_l}$

$\qquad$ Compute plug-in estimates of $\mu_{j,c}^l$ $\mu_{j,\bar{c}}^l$, $\Sigma_{j,c}^l$, $\Sigma_{j,\bar{c}}^l$ for all $j, c$

$\qquad$ Compute

$$r_j^l = \min_c \min_{\mu_{j,c}^l, \mu_{j,\bar{c}}^l} (\mu_{j,c}^l - \mu_{j,\bar{c}}^l)^\top (\Sigma_{j,c}^l + \Sigma_{j,\bar{c}}^l + (\rho_{j,\bar{c}}^l + \rho_{j,c}^l)I)^{-1}(\mu_{j,c}^l - \mu_{j,\bar{c}}^l)$$

$$\text{s.t. } \|\mu_{j,\bar{c}}^l - \bar{\mu}_{j,\bar{c}}^l\| \le \delta_{j,\bar{c}}^l$$

$$\|\mu_{j,c}^l - \bar{\mu}_{j,c}^l\| \le \delta_{j,c}^l$$

$\qquad$ for all $j$.

$\qquad$ **if** $j \in \mathrm{sort}_{B_l}(\{r_j^l\}_{j=1}^{N_l})$ **then**

$\qquad\qquad$ Set $s_j^l = 1$

**Output:** Sparse Masks $S_1, \cdots, S^L$

**return** $\hat{\mathcal{W}}$

---

## D.5 Recovering TVSPrune

In TVSPrune, at any layer $l$, the $j$th filter is pruned if

$$1 - e^{-\Delta_{l,j}}$$

where $\Delta_{l,j}$ is the minimum Fisher discriminant between pairs of classes. Suppose that we identify important and discriminative filters by measuring the TV distance in a pairwise sense. Recall that Corollary 2 gives us a bound that is also monotonic in $\Delta_{l,j}$. If we apply the strategy that we prune all filters with a score less than a threshold, we would prune filter $j$ if

$$\frac{\Delta}{2 + \Delta} \le \gamma$$

for some $\gamma \in (0,1)$. We can now find a relation between $\gamma$ and $\eta$. First, note that if $1 - e^{\frac{-\Delta}{4}} \leq \eta$, then $\Delta \leq 4(1-\eta)$. Similarly, if we prune $\frac{\Delta}{2+\Delta} \leq \gamma$, then $\Delta \leq \frac{2\gamma}{1-\gamma}$. Equating the two gives us the expression

$$\eta = \frac{3\gamma - 2}{2\gamma - 2}.$$

Thus, both TVSPrune and a variant of DISCEDIT-SP-F where the TV distance is measured pairwise, and which prunes at a threshold are equivalent, as they require pruning the $j$th filter if

$$\Delta \leq 4 - 4\eta = \frac{3\gamma - 2}{2\gamma - 2}.$$

### D.6 Using the BatchNorm Random Variables

The BatchNorm random variables for this layer are given by

$$\mathsf{BN}^l(X) = \left[\mathbf{1}^\top Y_1^l(X), \cdots, \mathbf{1}^\top Y_{N_l}^l(X)\right] = \left[\mathsf{BN}_1^l(X), \cdots, \mathsf{BN}_{N_l}^l(X)\right]. \tag{18}$$

As stated earlier, our goal is to minimize the TV distance between the distributions of the pruned and unpruned features; we use the BatchNorm random variables as a proxy for the features $Y^l(X)$. Next, the moments of $\mathsf{BN}^l(X)$ are given by

$$\mathbb{E}_{X \sim \mathcal{D}}\left[\mathsf{BN}_i^l(X)\right] = \mathsf{BN}_i^l = \left[\mu_i^l\right] \quad \text{and} \quad \mathrm{Var}(\mathsf{BN}_i^l(X)) = (\sigma_i^l)^2. \tag{19}$$

Suppose $\mathsf{BN}^l(X)$ is drawn from the distribution $\mathcal{D}_j^{\mathsf{BN},l}$, $\mathcal{D}_{j,c}^{\mathsf{BN},l}$ be the $c$th class conditional distribution, and let $\mathcal{D}_{j,\bar{c}}^{\mathsf{BN},l}$ be the distribution of features sampled from the complement of class $c$.

## E  Additional Experiments

In this section, we detail additional experiments conducted in the course of this investigation. In particular, we provide experimental results highlighting the utility of our lower bound on synthetic datasets, and provide expanded results for the Gaussianity tests provided in Section 6.1.

### E.1  Measuring the TV Distance between Linearly Inseparable Data

In this section, we use the bounds proposed in Corollary 2 to bound the TV distance between poorly separated datasets. We choose the 'Two Spirals' Dataset, and a dataset consisting of two zero-mean Gaussians with different variances. We choose $f(x) = u^\top(\varphi(X) - (\bar{\varphi}_1 - \bar{\varphi}_0)/2)$, where $\phi(X)$, when chosen to be of degree $d \geq 1$, is given by $\varphi(X) = 1 + (\mathbf{1}^\top X) + \cdots + (\mathbf{1}^\top X)^d$. We present our results in Figure 2. We observe that using the lower bound for Fisher outperforms the MPM lower bound on both toy datasets. However, the choice of polynomial witness functions clearly outperforms lower degree choices, particularly in the 'Two Gaussians' case.

### E.2  Effective Pruning of Hard-to-Prune Layers

In this section, we utilize the lower bounds provided in this paper to prune hard-to-prune layers in neural networks. As noted in Murti et al. [39], Liebenwein et al. [34], some layers, in particular the initial layers in the case of VGG-nets, are difficult to effectively sparsify. In this set of experiments, we aim to show that using the lower bounds proposed in this work, we are able to better identify discriminative filters in hard-to-prune layers, and therefore prune those layers more effectively.
**Experiment Setup:** We select a VGG16 model trained on CIFAR10. We fix pruning budgets of $40\%, 50\%, 60\%, 70\%, 80\%$. For each model, we then prune three hard-to-prune layers in isolation, and measure the impact on accuracy. We compare the following methods:
**TVSPrune:** We modify TVSPrune to prune a fixed budget, and using the BatchNorm random variables as described in Appendix D.
**DISCEDIT-SP-EQ:** We apply Algorithm 5 as presented in Appendix D using the features $\varphi(\mathbf{1}^\top X) = [\mathbf{1}^\top X, (\mathbf{1}^\top X)^2]$. Thus, for each $c$, $f_{(c)}(X) = u^\top(\varphi(X) - (\bar{\varphi}_c + \bar{\varphi}_{\bar{c}})/2))$.

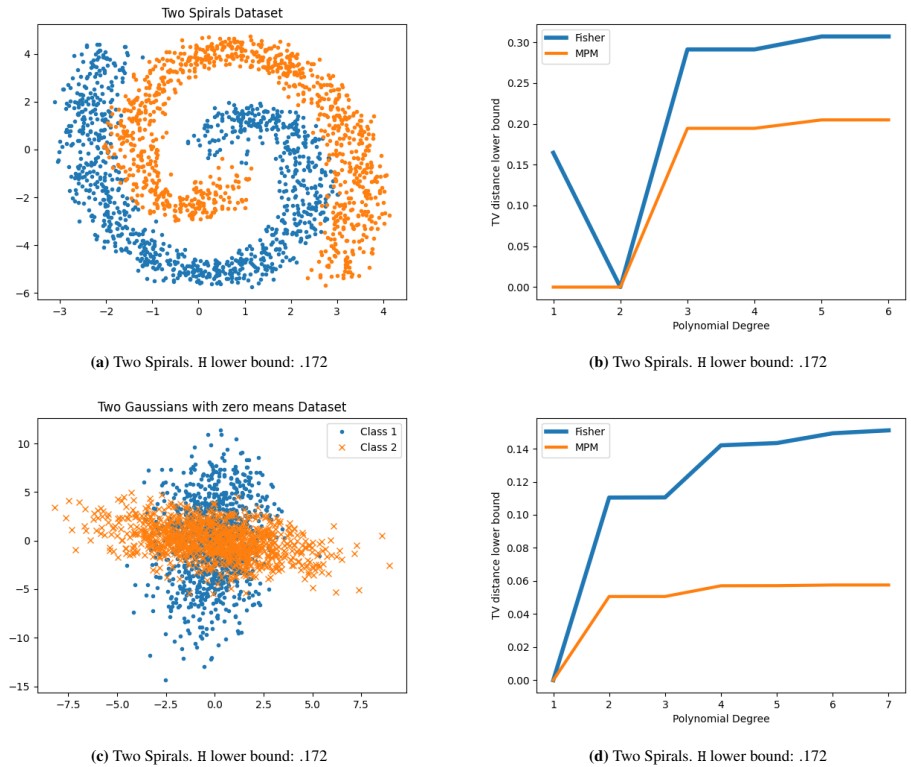

**(a)** Two Spirals. H lower bound: .172

**(b)** Two Spirals. H lower bound: .172

**(c)** Two Spirals. H lower bound: .172

**(d)** Two Spirals. H lower bound: .172

**Figure 2:** Comparison of the performance of DISCEDIT-SP F with DISCEDIT-SP M with polynomial features on the TwoSpirals and Zero-Means Gaussians datasets.

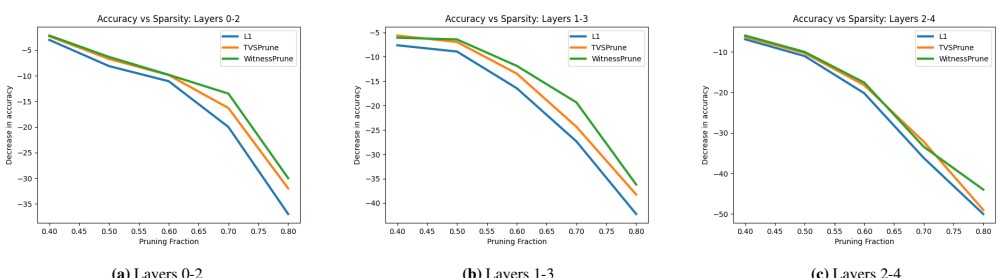

**(a)** Layers 0-2

**(b)** Layers 1-3

**(c)** Layers 2-4

**Figure 3:** Comparison of the performance of DISCEDIT-SP F with TVSPrune and L1 pruning on hard-to-prune layers in VGG16 trained on CIFAR10

$L_1$**-based Pruning:** We use the $L_1$ norms of the filter weights, as proposed in [32].

**Results and Discussion** We present our results in Figure 3 The experiments show that DISCEDIT-SP variants using quadratic features (using algorithms outperform both TVSPrune and the $L_1$-norm-based pruning strategy. In particular, we see that at 70% sparsity in Layers 1-3, the models obtained by DISCEDIT-SP-EQ are 6.6% more accurate than those obtained using TVSPrune.

### E.3 Verifying class-conditional Feature Maps are not Gaussian

In this section, we attempt to validate the assumptions made in Murti et al. [39] about the normality of the class-conditional feature distributions. To do so, we apply the Shapiro-Wilks test [49], a standard test for Normality.

**Experiment Setup:** We consider a VGG16 model trained on CIFAR10. Let $\mathsf{BN}_j^l(X) = 1^\top Y_j^l(X)$. For each $l, j$, we collect 100 samples from each class $c \in [10]$. We then apply the Shapiro-Wilk normality test [49], and we compute $p_{j,c}^l$ values, which are the minimum $p$-values from the Shapiro-Wilks test computed for the features of the $j$th filter in layer $l$ conditioned on class $c$. We consider that a

filter's features are unlikely to be Gaussian if $p_{j,c}^l < 0.1$. We plot the heatmaps of $p_j^l = \min_{c \in [C]} p_{j,c}^l$ values for 15 randomly selected filters in Figure 4, to indicate the normality of the least Gaussian class-conditional features.

**Results:** We observe that for most layers, particularly those close to the output, the class-conditional feature distributions are highly unlikely to have been drawn from a Gaussian, with $p_{j,c}^l$ values for layers 10-12 in VGG16 typically being below $1e-5$. For layers with filters that yield likely-Gaussian features, we observe that for the majority of filters, at least one feature output is likely to be non-Gaussian. We present this data in Figure 4.

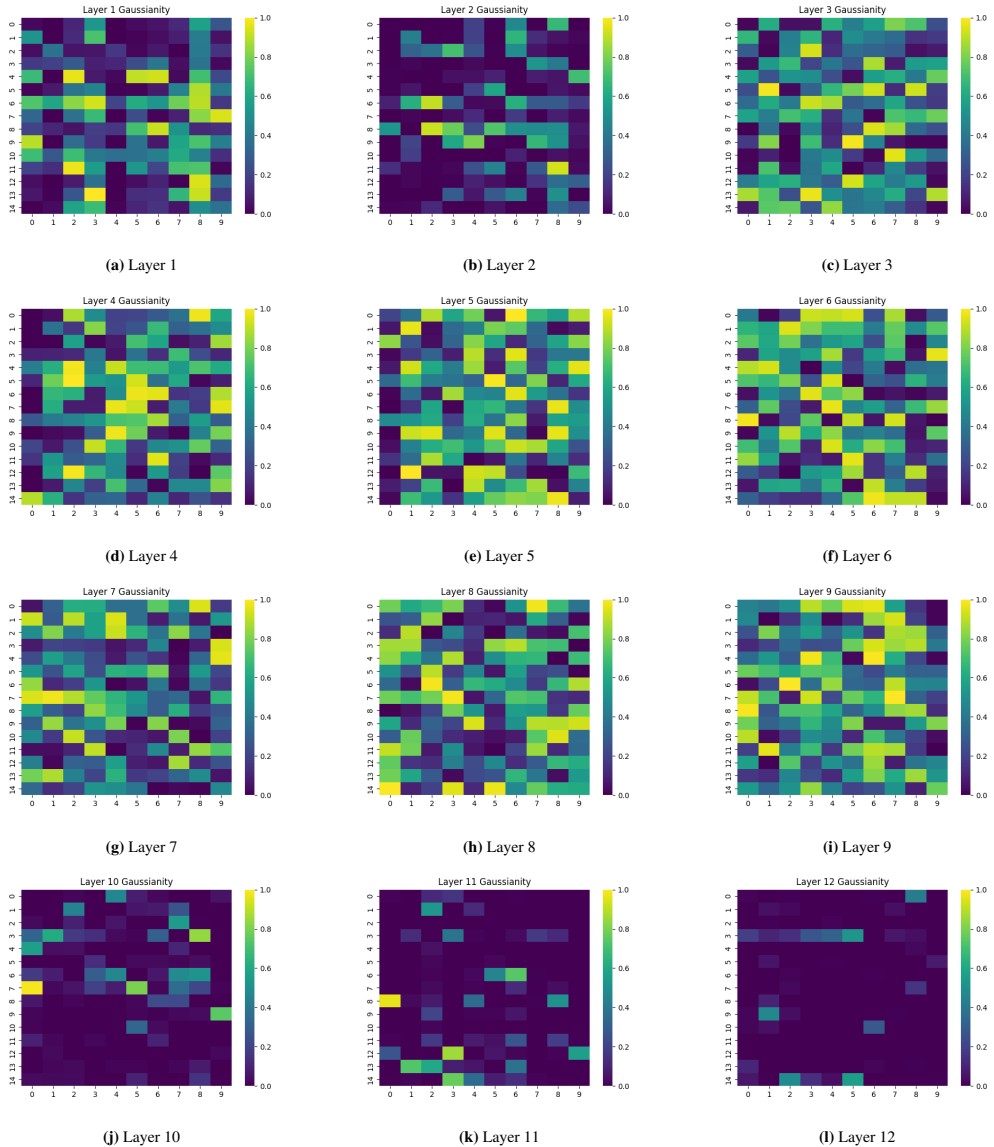

**Figure 4:** Shapley-Wilks test $p$-value heatmaps applied to class-conditional features generated by filters in each layer. $x$-axis is the class index, and $y$-axis is the filter index. Filters shown in the heatmap are selected randomly.

## E.4 Ablation Experiments for DISCEDIT-SP

In this section, we conduct a slate of experiments aimed at demonstrating that the effectiveness of DISCEDIT-SP is not dependent on particular instantiations of models. Specifically, we apply DISCEDIT-SP to multiple instances of models trained on CIFAR10, CIFAR100, and Imagenet. We present specific results in the subsequent subsections.

### E.4.1 CIFAR10 Experiments

In this section, we run DISCEDIT-SP on 10 different instantiations of different models trained on CIFAR10. We observe that DISCEDIT-SP performs well on all instances of the models, irrespective of architecture. Our results are tabulated in Table 4.

**Table 4:** Ablations for models trained on CIFAR10. We consider 10 different models, and average the accuracy after pruning/fine-tuning over them.

| Model | Sparsity | Best Acc. drop | Worst Acc. Drop | Mean Acc. Drop |
|-------|----------|----------------|-----------------|----------------|
| VGG16 | 40.8% | +0.32 | −0.06 | +0.09 ± 0.06 |
| VGG16 | 61.2% | 0.19 | 0.51 | 0.35 ± 0.03 |
| VGG16 | 75.6% | 1.27 | 1.38 | 1.34 ± 0.01 |
| ResNet56 | 41.2% | +0.03 | 0.05 | 0.01 ± 0.01. |
| ResNet56 | 60.7% | 1.21 | 1.30 | 1.24 ± 0.02 |

### E.4.2 CIFAR100 Experiments

In this section, we run our experiments on 5 different instances of VGG16, VGG19, and ResNet56 models trained on CIFAR10. We present our results in Table 5. Our results show that the effectiveness of DISCEDIT-SP is unrelated to the particular instantiation of the model.

**Table 5:** Ablations for models trained on CIFAR100. We consider 10 different models, and average the accuracy of our pruning algorithm over them.

| Model | Sparsity | Best Acc. drop | Worst Acc. Drop | Mean Acc. Drop |
|-------|----------|----------------|-----------------|----------------|
| VGG16 | 40.8% | +0.03 | 0.06 | 0.03 ± 0.06 |
| VGG19 | 60.6% | 0.12 | 0.19 | 0.16 ± 0.03 |
| Resnet56 | 40.2 | +0.13 | 0.16 | ±0.01 |

### E.4.3 Imagenet Experiments

In this section, we provide ablation experiments for Imagenet. However, owing to the computational cost of training Imagenet models, we only train 2 additional instantiations of the model. Moreover, we were only able to fine-tune for 30 epochs. However, DISCEDIT-SP broadly works well on all of the instantiations, as seen in Table 6.

**Table 6:** Ablations for models trained on Imagenet. We consider 3 different models, and average the accuracy drop.

| Best Accuracy | Worst Accuracy Drop | Mean accuracy drop |
|---------------|---------------------|--------------------|
| 3.1% | 8.3% | 6.6% |

### E.4.4 Pruning without Fine-tuning

In this section, we highlight our pruning experiments without fine-tuning models. We use fixed layerwise sparsity budgets. We use models trained on CIFAR10 and Imagenet datasets. Our experiments, tabulated in Table 7, shows that our method consistently matches or outperforms common baselines.

### E.5 Pruning in the High Sparsity Regime

In this section, we prune models trained on CIFAR10 extensively, with over 80% of parameters pruned in total. Fine-tuning is done in a one-shot fashion. We see that DISCEDIT-SP consistently matches our outperforms baselines such as Murti et al. [39] or Sui et al. [52]. Our results are tabulated in Table 8

**Table 7:** Pruning results with DISCEDIT-SPwithout fine-tuning.

| Model | Dataset | Sparsity | Acc. Drop (ours) | Acc. Drop [39] | Acc. Drop [52] | Acc. Drop (L1) |
|---|---|---|---|---|---|---|
| VGG16 | CIFAR10 | 32% | 1.91 | 2.20 | 2.13 | 10.52 |
| VGG16 | CIFAR10 | 41% | 4.56 | 5.21 | 6.53 | 16.59 |
| VGG16 | CIFAR10 | 63% | 10.16 | 12.79 | 12.65 | 32.8 |
| VGG19 | CIFAR10 | 30% | 0.98 | - | 1.22 | 6.55 |
| VGG19 | CIFAR10 | 44% | 2.56 | - | 4.53 | 13.8 |
| VGG19 | CIFAR10 | 60% | 6.16 | - | 8.44 | 23.67 |
| ResNet50 | ImageNet | 11% | 18.37 | 22.61 | 21.49 | 35.52 |
| ResNet50 | ImageNet | 21% | 30.7 | 36.0 | 32.69 | - |

**Table 8:** Pruning results with DISCEDIT-SP on CIFAR10 models in the high-sparsity regime

| Model | Sparsity | Algorithm | Acc. drop (no FT) | Acc. drop (FT) |
|---|---|---|---|---|
| | 84.7% | CHIP | 83.1% | 0.20% |
| VGG16 | 84.7% | TVSPrune | 83.4% | 0.21% |
| | 84.7% | DISCEDIT-SP | 82.9% | 0.15% |
| | - | CHIP | - | - |
| | 88.7% | TVSPrune | 84.8% | 0.17% |
| VGG19 | 88.7% | DISCEDIT-SP | 84.7% | 0.16% |
| | 74.3% | CHIP | 84.0% | 0.28% |
| ResNet56 | 74.3% | TVSPrune | 83.9% | 0.25% |
| | 74.3% | DISCEDIT-SP | 84.2% | 0.28% |

## E.6 Discriminative Component Discovery for CIFAR10 models

In this section, we provide plots of $\eta_{l,j}^c$ values for different layers in models trained on CIFAR10. We show plots for 3 different layers for 4 classes, to highlight how discriminative components look.

### E.6.1 ResNet56 trained on CIFAR10

In this section, we produce plots showing discriminative and nondiscriminative components for a ResNet56 trained on CIFAR10. Our results are plotted in Figure 5. Note that in layer 0, both class 0 and class 8 have filter 8 as discriminative. Moreover, in subsequent layers, all three classes have filters which are somewhat discriminative as well.

### E.6.2 VGG16 trained on CIFAR10

Similar to the previous subsection, we plot $\eta_{l,j}^c$ values for a VGG16 trained on CIFAR10 We plot a selection of our results in Figure 6. As before, we see that we can clearly identify discriminative and nondiscriminative filters. Note again that class 0 and class 8 share discriminative filters in both layer 0 as well as in layer 4.

## F  Additional Experimental Details

In this section, we detail additional experiments not mentioned in the main paper, as well as a comprehensive description of our experimental setup.

### F.1  Pruning Setup

In this section, we discuss our experimental setup.

### F.1.1  Platform Details

The hardware used for the experiments in this work are detailed below:

1. Server computer with 2 NVIDIA RTX3090Ti GPUs with Intel i9-12700 processors, running Ubuntu 20.04, with Python 3.11 and CUDA Tools 10.2 with PyTorch 2.0.1.

2. Desktop computer with 1 NVIDIA RTX3070Ti GPUs with Intel i7-10700 processor, running Ubuntu 22.04, with Python 3.11 and CUDA Tools 11.7 with PyTorch 2.0.1.

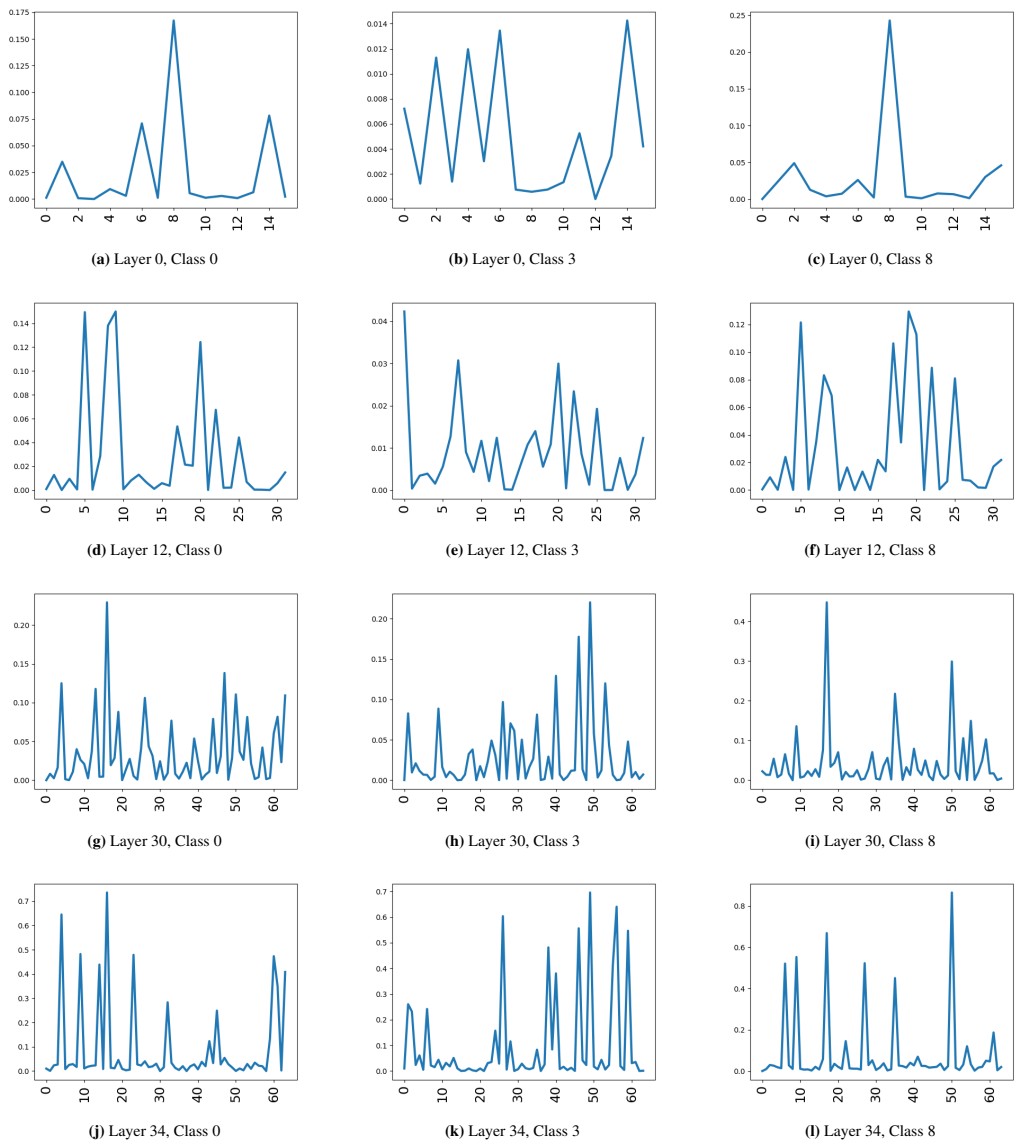

**Figure 5:** $r_{l,j}^c$ plots for different layers, and classes 0, 3, and 8 for a ResNet56 trained on CIFAR10. $x$ axis is the filter index, $y$-axis is $r_{l,j}^c$.

#### F.1.2 Models under consideration

We consider the following models.

- **VGG16/19 trained on CIFAR10 and CIFAR100:** We use the pre-trained VGG11/16/19 models trained on CIFAR10 and CIFAR100. The models achieve accuracies greater than 90% on both datasets.
- **ResNet56 trained on CIFAR10:** We consider a ResNet56 model trained on CIFAR10. We do not prune layers that are part of complex interconnections (such as the final layer in each BasicBlock).
- **ResNet50 trained in Imagenet:** We consider a ResNet50 model trained on Imagenet. We do not prune layers that are part of complex interconnections, as was the case in ResNet56.
- **ViT trained on CIFAR10** We trained a custom ViT on CIFAR10. The details of the model are given below in Table 9: The accuracies of this models are given in Table 10

The accuracies of all models are given in Table 11 below:

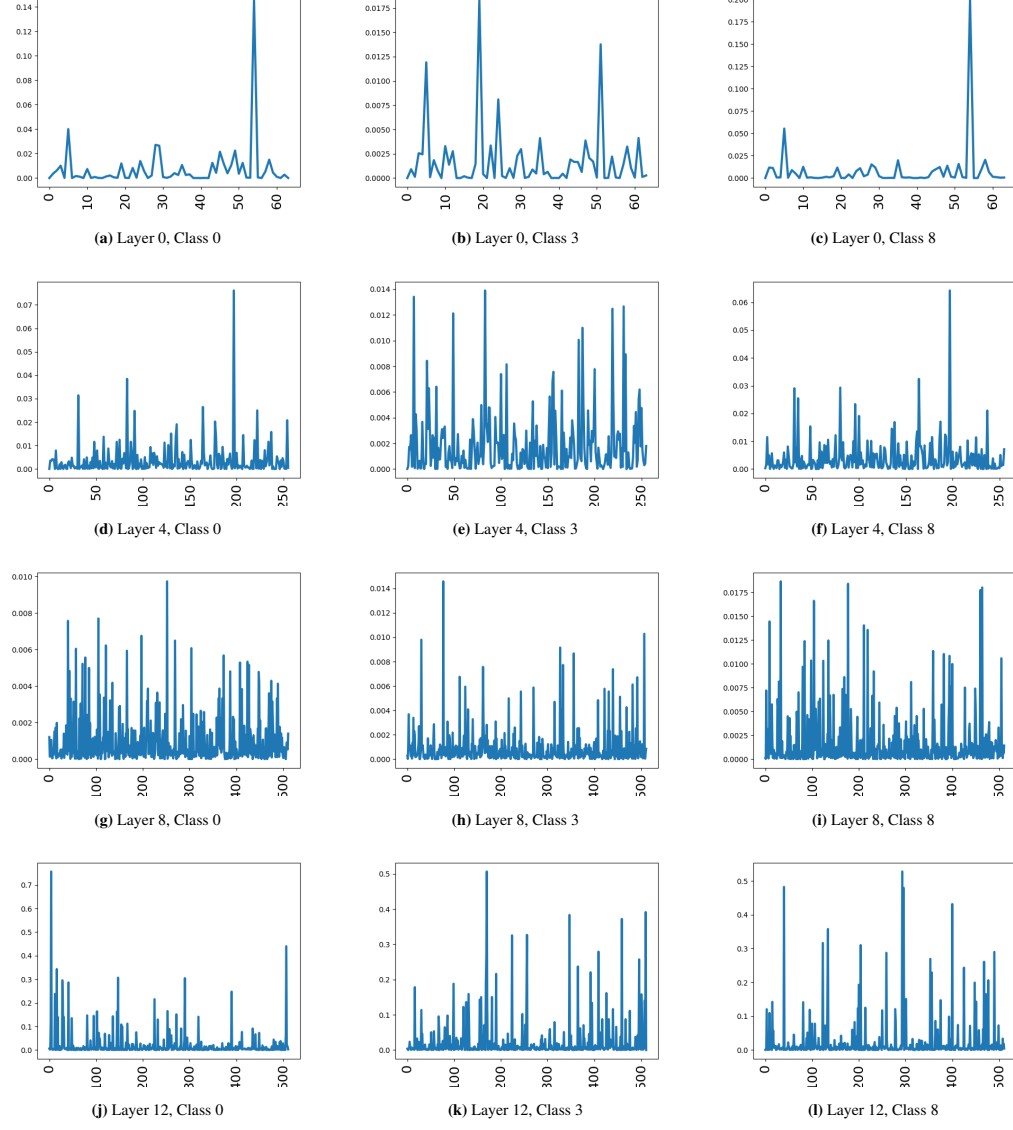

**Figure 6:** $r_{l,j}^c$ plots for different layers, and classes 0, 3, and 8 for a VGG16 trained on CIFAR10. $x$ axis is the filter index, $y$-axis is $r_{l,j}^c$.

| Parameter | Value |
|---|---|
| Context Length | 65 |
| Embedding dim. | 384 |
| Transformer Encoders | 7 |
| MLP layers to be pruned | 14 |
| Total params. | 6.27m |

**Table 9:** Summary of model parameters.

| Dataset | Test Acc. |
|---|---|
| CIFAR10 | 88.2% |
| CIFAR100 | 69.5% |

**Table 10:** Test accuracy for CIFAR datasets.

| Dataset | Model | Test Accuracy |
|---|---|---|
| CIFAR10 | VGG16 | 94.16 |
| | ResNet56 | 94.37 |
| | ResNet20 | 92.2 |
| | ViT | 88.2 |
| CIFAR100 | VGG16 | 74.0 |
| | ResNet56 | 72.6 |
| | ViT | 69.5 |
| ImageNet | ResNet50 | 76.15 |

**Table 11:** Test accuracy for different datasets and models.

**Model Provenance**    We list the sources of the models below.

- All CIFAR10 and CIFAR100 models were obtained from:
  `https://github.com/chenyaofo/pytorch-cifar-models`.

- ResNet50 trained on Imagenet was obtained from:
  `https://drive.google.com/drive/folders/1b-dZlvKUUu0rXqMYAtIr0ynHQHuEWDI`,
  which in turn comes from:
  `https://github.com/Eclipsess/CHIP_NeurIPS2021?tab=readme-ov-file`

- The ViT models use code from `https://github.com/omihub777/ViT-CIFAR`.

### F.1.3    Dataset Selection

**For Pruning**    Since we assume that the training dataset is unavailable to us, we utilize the validation set as a proxy for the data-distribution. We detail our dataset splits in Table 12. **Note: the subset**

**Table 12:** Breakdown of dataset splits used in our experiments.

| Dataset | Training Set | TV Distance Set | Test Set |
|---|---|---|---|
| CIFAR10 | Not used | 4000 images from test set | 6000 images from Test set |
| CIFAR10 | Not used | 4000 images from test set | 6000 images from Test set |
| Imagenet | Not used | 30000 images from Validation set | 20000 images from Val. set |

**used to compute the TV distances are not reused while measuring the test accuracy.**

**Class Unlearning**    We use the training set to identify discriminative filters, and the test set to measure accuracy. Typically, we get

### F.1.4    Hyperparameter Details

In this section, we detail the hyperparameters used when fine-tuning pruned models. We present the hyperparameters for CIFAR10 and Imagenet models only, as we did not fine-tune models that used CIFAR100.

**CIFAR10 Fine-tuning**    We detail the hyperparameters used in our CIFAR10 experiments below.

1. **Batch Size:** 128

2. **Epochs:** 50

3. **Learning Rate:** .001

4. **Weight Decay:** .0005

5. **Momentum paramters:** .9

6. **Optimizer:** SGD

**ImageNet Fine-tuning** We detail the hyperparameters used when fine-tuning Imagenet models.

1. **Batch Size:** 128, using gradient accumulation
2. **Epochs:** 100
3. **Learning Rate:** 0.08 (initial)
4. **Momentum:** 0.99
5. **Weight Decay:0.0001**
6. **Optimizer:** SGD

