# OpenReview forum: "DisCEdit: Model Editing by Identifying Discriminative Components"
_NeurIPS.cc/2024/Conference — NeurIPS 2024 poster_

### Official Review · Reviewer_crkh · 2024-07-08

**Soundness:** 3
**Presentation:** 2
**Contribution:** 2
**Rating:** 5
**Confidence:** 3

**Summary:**

This paper applies model editing to address two active areas of research, Structured Pruning and Selective Class Forgetting.
Specifically, it adopts a distributional approach to identify important components useful to the model's predictions.
With the witness function-based lower bounds on the TV distance, it is able to discover critical subnetworks responsible for classwise predictions, thus achieving Structured Pruning and Selective Class Forgetting.

**Strengths:**

1. This paper uses lower bounds on the TV distance to approximate the Bayes error rate to quantify the discriminative ability of a filter in a neural network.

2. This paper introduces DISCEDIT-U, which selectively prunes components capable of discriminating each class to facilitate unlearning and DISCEDIT-SP, which prunes non-discriminative filters to achieve structured pruning.

3. The experimental results demonstrate the efficacy of DISCEDIT in both unlearning and structured pruning.

**Weaknesses:**

1. The code provided by the authors redirects to an empty GitHub repository, lacking reproducibility.

2. This paper focuses on the well-established research areas of machine unlearning and structured pruning.
However, the authors fail to provide an introduction to the field or an overview of the current state of existing research.

3. In the experiments of DISCEDIT-U, the authors do not compare its results with other existing unlearning methods, making it difficult to ascertain whether it outperforms the current baselines.

4. The experimental results of DISCEDIT-SP show no significant improvement over the compared baselines, especially when evaluated against CHIP on ImageNet.

**Questions:**

1. The authors should provide a more detailed description of machine unlearning and structured pruning, along with a comprehensive introduction to the current research landscape in these areas.

2. To demonstrate the effectiveness of the proposed DISCEDIT-U, the authors should compare its performance against other works within the field.

3. The authors should incorporate a more detailed introduction of the experimental setup and the results into the main text, rather than relegating them to the appendix without sufficient description.

---

> ### Author Rebuttal · Authors · 2024-07-31
>
> We thank the reviewer for the constructive feedback and useful suggestions.
>
> ### Weaknesses:
>
> **The code provided by the authors redirects to an empty GitHub repository, lacking reproducibility.**
>
> We wish to immediately amend the issue with the empty github and apologize for the mistake. We provide a corrected link below:
>
> https://rb.gy/9zaxzz
>
> **This paper focuses on the well-established research areas of machine unlearning and structured pruning. However, the authors fail to provide an introduction to the field or an overview of the current state of existing research.**
>
> We provide a detailed related works section in Appendix A, lines 538-598.   Moreover, in lines 30-38 and section 3, we introduce the problem of model editing, particularly by using component attribution.
>
> **In the experiments of DISCEDIT-U, the authors do not compare its results with other existing unlearning methods, making it difficult to ascertain whether it outperforms the current baselines.**
>
> We thank the reviewer for the suggestion, and have compared our results with additional baselines in the General response.
>
> **Takeaway: our proposed method for model unlearning is competitive with current state of the art without requiring additional retraining, simply by identifying and masking discriminative components.**
>
> **The experimental results of DISCEDIT-SP show no significant improvement over the compared baselines, especially when evaluated against CHIP on ImageNet.**
>
>  In the work, we show that with extensive fine-tuning, all models recover nearly the entire accuracy of the original model. However,  in Table 7 of the Appendix, *we present results for pruning without fine-tuning*. Our results clearly outperform near baselines such as CHIP [1] and TVSPrune [2].
>
>  **Takeaway: We show that our proposed method outperforms the nearest baselines in this regime (without extensive fine-tuning).**
>
> ### Questions:
>
> **The authors should provide a more detailed description of machine unlearning and structured pruning, along with a comprehensive introduction to the current research landscape in these areas.**
>
> The problems of structured pruning and classwise unlearning are stated in Section 3 in the paper, on lines  145-163. However, we formally state the problems of machine unlearning and structured pruning in the sequel. Should the work be accepted, we will incorporate these problem definitions into Section 3, where we set up the two problems. Moreover, a detailed discussion of related work is presented in Appendix A.
>
> **Machine Unlearning**
> Let $\mathcal{D}$ be the data distribution, let $\mathcal{D}\_{c}$ be the distribution of class $c$ in the dataset, and let $\mathcal{D}\_{\bar{c}}=  \mathcal{D}\backslash \mathcal{D}\_c$ be the distribution of the remaining classes. Let $f\_\theta(X)$ be a neural network with parameters $\theta$ trained on samples drawn from $\mathcal{D}$, with loss function $\mathcal{L}(\cdot)$. Unlearning class $c$ by editing $\theta$ can be formalized as finding parameters $\theta^*$, given some $\epsilon > 0$, using:
>
> $\theta^* = \arg\max\_{\theta'} \mathbb{E}\_{X\sim\mathcal{D}\_c}\mathcal{L}(f\_{\theta'}(X))\quad \text{s.t. }|\mathbb{E}\_{X\sim\mathcal{D}\_{\bar{c}}}[\mathcal{L}(f\_{\theta'}(X))] - \mathbb{E}\_{X\sim\mathcal{D}\_{\bar{c}}}[\mathcal{L}(f\_{\theta}(X))]|\leq \epsilon$
>
> That is, we edit the parameters to maximize the loss on $\mathcal{D}\_c$ while minimizing the effect on $\mathcal{D}\_{\bar{c}}$.
>
> Similarly, structured pruning with a fixed budget $K$ can be written as a search for sparse parameters $\theta^*$
> such that
>
> $\theta^* = \arg\min\_{\theta'} \mathbb{E}\_{X\sim\mathcal{D}}[\mathcal{L}(f\_{\theta'}(X)] \text{ s.t. } ||\theta'||_0\leq K$
>
> Thus, we aim to find a sparse set of parameters (with at most $K$ nonzero parameters) that minimizes the loss.
>
> **To demonstrate the effectiveness of the proposed DISCEDIT-U, the authors should compare its performance against other works within the field.**
>
> We thank the reviewer for the suggestion and present a comparison with additional baselines in the General Response.
>
> **Takeaway: Our proposed methods are competitive with the additional baselines without requiring additional retraining/fine-tuning.**
>
> **The authors should incorporate a more detailed introduction of the experimental setup and the results into the main text, rather than relegating them to the appendix without sufficient description.**
>
> Should this work be accepted, we will add a more detailed description of the experimental setup to the main body of the work.

---

> ### Author Response · Authors · 2024-08-11
> **Response to Reviewer**
>
> We have responded to the concerns raised by the reviewer in our previous response. In particular,  we have
> - addressed the empty github link
> - highlighted our related works section in Appendix A
> - shown the comparison of our models as compared to baselines *without fine-tuning*
> - given formalizations of unlearning and pruning in addition to what we presented in Section 3 of the main document.
> - Comparisons with baselines for model unlearning have been presented in the general rebuttal [**here**](https://openreview.net/forum?id=tuiqq1G8I5&noteId=8sBlgK4EB5).
> - Moreover, we have added the requested ViT experiments (by reviewers YbA1 and FULk) as a response to the General Rebuttal [**here**](https://openreview.net/forum?id=tuiqq1G8I5&noteId=MiXju1ozmG).
>
> We hope that we have adequately addressed your concerns, and are eager to find out what we might do to improve your appraisal of our work.  We would like to gently remind the reviewer that the author-reviewer discussion period ends in 3 days, and we are eager to engage further with you.

---

> ### Author Response · Authors · 2024-08-12
> **Gentle Reminder about approaching Auhor-Reviewer discussion deadline**
>
> We'd like to provide a gentle reminder that the author-reviewer discussion period ends in 36 hours. We hope that we've addressed the concerns raised in your review in our previous responses, and we are very eager to engage with you further if you have additional concerns, and what we might do to improve your evaluation of our work.

---

### Official Review · Reviewer_FULk · 2024-07-12

**Soundness:** 3
**Presentation:** 2
**Contribution:** 3
**Rating:** 7
**Confidence:** 3

**Summary:**

In this work, the authors propose to tackle two problems at once, class unlearning and structured pruning. To do so, they propose a novel way to compute a lower bound on the total variation distance between distributions of features.
In practice, this distance is approximated using the first and second order moments of a transformation of intermediate features generated from samples of specific classes. The aforementioned transformations are specific to the task of class unlearning or structured pruning.
The resulting method comes with a strong theoretical background, completed by convincing empirical evidence regarding.

**Strengths:**

This work bridges the gap between, selective class forgetting and efficient inference. In its approach, this work is quite original and generalizes previous methods by removing questionable assumptions.
The presentation of the result is overall clear, despite the strong mathematical grounding of the proposed method.
The empirical results show that the proposed technique slightly improves over previous similar approaches.

**Weaknesses:**

I have one minor concern regarding the evaluation of the method:
In the main paper, the authors insist on the ability of the proposed method to better discriminate important filters in the case of pruning and selective class forgetting. However, the experiments all include fine-tuning and a quite extensive one on pruning with up to 100 epochs. In my understanding, this empirically validates that the filter selection leads to a selection that can be better fine-tuned, not necessarily that the selection alone better preserves the performance.
I think that some experiments in that regard were conducted and discussed in the appendices, but they should be moved to the main paper if possible. Also, they should be extended to ImageNet, also, if possible.

**Questions:**

I have three questions:
1. (main point from weaknesses) can the author provide results on ImageNet without fine-tuning to highlight the better selection of filters from their method in the main paper?
2. It is now standard to include transformers when evaluating pruning techniques. If the authors could provide results on Bert, that would benefit the scope of the paper.
3. In the article, it is claimed that the method can perform structured pruning with no access to the data. However, from my understanding, the saliency score from equation 9 uses Y(X) which is computed with some data. Could the authors elaborate on this point, please?

**Limitations:**

The authors have adequately addressed the limitations of their work.

---

> ### Author Rebuttal · Authors · 2024-08-07
>
> We thank the reviewer for the positive appraisal of our work, and
> ### Questions:
>
> **(main point from weaknesses) can the author provide results on ImageNet without fine-tuning to highlight the better selection of filters from their method in the main paper?**
>
> We thank the reviewer for the insightful question. We have provided results for pruning ImageNet and CIFAR10 models without fine-tuning in Table 7 of the Appendix.  In that table, we we show that **our structured pruning algorithm, DisCEdit-SP, outperforms baselines to a greater extent in the regime without fine-tuning.** Note that at higher sparsity regimes, the accuracy of Imagenet models falls to less than 1% on all baselines.
>
>
> **It is now standard to include transformers when evaluating pruning techniques. If the authors could provide results on Bert, that would benefit the scope of the paper.**
>
> We thank the reviewer for the suggestion. We will present results on ViTs trained on CIFAR10 for both unlearning and pruning (without fine-tuning) shortly.
>
> **In the article, it is claimed that the method can perform structured pruning with no access to the data. However, from my understanding, the saliency score from equation 9 uses Y(X) which is computed with some data. Could the authors elaborate on this point, please?**
>
> As stated in lines 83-84 in Section 1, motivated by recent works highlighting the need to be able to compress models without training data or the loss function, our proposed method requires no access to *original training data (data upon which the model was trained)*; rather, we only require distributional access, either via samples from the original distribution not in the training set, or in the form of *finitely many moments*, which can be used in conjunction with Theorem 2 to identify discriminative components.

---

> > ### Comment · Reviewer_FULk · 2024-08-08
> > **Response to rebuttal**
> >
> > I would like to thank the authors for their response. I would recommend moving Table 7 from the appendix to the main paper since it appears to answer many concerns from several reviewers. I consider that most of my points have been addressed, and I am looking forward to seeing your results on ViT.

---

> ### Author Response · Authors · 2024-08-11
> **Response to Reviewer**
>
> We thank the Reviewer for engaging with us so clearly early on, as well as the patience shown! We hope that this adequately addresses the reviewer's concerns.
>
> As regards Table 7, we thank you for the suggestion, and will incorporate Table 7 into the main document
>
> We have presented the requested experiments on ViTs as a response to the general Rebuttal [**here**](https://openreview.net/forum?id=tuiqq1G8I5&noteId=MiXju1ozmG). Our experimental results highlight the fact that the proposed approaches for model editing (pruning and classwise unlearning) are effective when used with Vision Transformers as well.
>
> As the author-reviewer discussion period ends in 3 days, we are eager to engage further and would like to know what we might do to improve the reviewer's appraisal of our work.

---

> > ### Author Response · Authors · 2024-08-12
> > **Gentle Reminder about Author-Reviewer Discussion Deadline**
> >
> > We'd like to gently remind the reviewer that the Author-Reviewer discussion period ends in 36 hours. We sincerely hope that we've addressed the concerns you expressed, and we're eager to engage further with you, and identify how we might improve your rating of our work.

---

### Official Review · Reviewer_H2n6 · 2024-07-13

**Soundness:** 3
**Presentation:** 4
**Contribution:** 3
**Rating:** 6
**Confidence:** 3

**Summary:**

The paper addresses the task of model editing that focuses on modifying critical components within neural networks to improve performance. One of the cornerstone steps is to first identify these components. The authors adopt an approach based on recently proposed discriminative filters hypothesis. Instead of using a Total Variation distance (which is intractable in this case), the authors derive a lower bound on the TV that is subsequently used to discover critical subnetworks responsible for classwise predictions.
The authors introduce algorithms for structured pruning and selective class forgetting and experimentally show its performance.

**Strengths:**

- The paper is well written, and the ideas are well motivated.
- The proposed method looks efficient. The problem that is being addressed is significant, so improvements in this area can potentially benefit the community.
- The authors provided a thorough analysis and theoretical justifications of the proposed method (including findings in Appendix).

**Weaknesses:**

- It is quite challenging to evaluate the performance of the propose solution due to the absence of comparison to other methods. While it is possible to see that the method is capable of preserving the overall test accuracy almost unchanged while reducing the quality of predictions on a chosen class by ~80%, it is still challenging to interpret these values. It looks to me that the methods listed in A.4 can be relevant and included into comparison.
- Related work is moved to Appendix. In my opinion, it would be better to include at least a shortened version into the main part of the paper so that the readers can easier understand the relations of the proposed method to the existing literature.

**Questions:**

NA

**Limitations:**

The authors has adequately addressed the limitations.

---

> ### Author Rebuttal · Authors · 2024-08-07
>
> We thank the reviewer for the positive appraisal of our work, and we address the concerns raised below.
>
> ### Weaknesses:
>
> **It is quite challenging to evaluate the performance of the propose solution due to the absence of comparison to other methods. While it is possible to see that the method is capable of preserving the overall test accuracy almost unchanged while reducing the quality of predictions on a chosen class by ~80%, it is still challenging to interpret these values. It looks to me that the methods listed in A.4 can be relevant and included into comparison.**
>
> We thank the reviewer for the suggestion, and provide a comparison with additional recent baselines in the general response. We refer the reader to the General Response for a summary of our experimental results.
>
> **Takeaway: Our work achieves comparable performance to current state of the art without requiring any fine-tuning or retraining, simply by identifying and editing discriminative filters.**
>
> **Related work is moved to Appendix. In my opinion, it would be better to include at least a shortened version into the main part of the paper so that the readers can easier understand the relations of the proposed method to the existing literature.**
>
> We thank the reviewer for the suggestion. Should the work be accepted, we will include a summary of the work presented in Appendix A  in the main document.

---

> ### Author Response · Authors · 2024-08-11
> **Response to Reviewer**
>
> We hope we have adequately addressed the reviewer's concerns, and thank you for your positive review of our work.
>
> In response to the concerns you raised, we would like to point out that we have:
> - provided comparisons with additional baselines in the general rebuttal [**here**](https://openreview.net/forum?id=tuiqq1G8I5&noteId=8sBlgK4EB5), as requested by the reviewer.
> - we will add at least a truncated version of the related works section into the main manuscript
> - We have presented experimental results on a simple ViT as asked by reviewers crkh, FULk, and YbA1 as an additional comment to the General Rebuttal [**here**](https://openreview.net/forum?id=tuiqq1G8I5&noteId=MiXju1ozmG).
>
>
> We would like to gently remind you that the author-reviewer discussion period ends in 3 days, and we are eager to engage further, and to find out what we might do to improve your appraisal of our work.

---

> > ### Comment · Reviewer_H2n6 · 2024-08-12
> > **Rebuttal**
> >
> > Thank you for your answers! I decided to keep my rating unchanged.

---

> > > ### Author Response · Authors · 2024-08-12
> > > **Thanks!**
> > >
> > > We thank the reviewer for the engagement! As the author-reviewer discussion period comes to a close in the next 36 hours, we're eager to continue to engage with you, particularly if there are any other unaddressed concerns you might have, and what we might do to improve your appraisal of our work.

---

### Official Review · Reviewer_YbA1 · 2024-07-17

**Soundness:** 4
**Presentation:** 3
**Contribution:** 3
**Rating:** 7
**Confidence:** 4

**Summary:**

This paper proposes a method for model editing of convolutional classifier networks. The proposed approach assess the class-discriminative ability of convolutional filters by looking at the distribution of their produced feature maps. Specifically, by comparing the class conditional feature distribution to the marginal feature distribution (which is related to the Bayes error rate of the filter), this work is able to produce a saliency score for each filter that determines its importance for classifying a specific class.  For model pruning, the least discriminative filters are pruned, whereas for class forgetting, the most discriminative filters for the forget class are removed.

This work involves several technical innovations; primarily,  they approximate computation of the TV distance between the class conditional and marginal feature distributions by proposing a moment-based lower bound, which does not make assumptions about the feature distributions, such as Gaussianity.

Numerical results reported for model pruning and class forgetting using ResNets and VGG nets on ImageNet, CIFAR10, CIFR100. Results demonstrate compelling performance. Interesting additional results in the appendix, e.g., analyzing the distribution of the feature maps.

The contribution of this work is both theoretical and empirical.

**Strengths:**

**Originality**
* This work is novel to the best of my knowledge. In particular, the main innovation in this work is the removal of the Gaussian feature distribution assumption from previous works by expending many pages to providing a moment-based lower bound on the TV distance, which can be estimated form few samples.

**Quality**
* This work is of high technical quality. Numerical results are compelling, demonstrating significant class forgetting with minimal model editing, and similarly, demonstrating strong performance even with significant pruning.

**Clarity**
* The paper is well written and relatively straightforward to follow, although some design choices; e.g., choice of witness function in numerical results are not clearly motivated.

**Significance**
* The problem of model editing is likely of broad interest to the neurips community, and the proposed approach is conceptually simple and does not require access to the loss function.

**Weaknesses:**

* Numerical results both in the main paper and the appendix do not report the actual accuracies of any of the models. Instead, they only report relative drops in performance on forget and non-forget classes. The link provided for the ImageNet classifier in line 910 in the appendix does not work.
* Computational complexity of the approach compared to related works is not clear; specifically, what is the cost of computing the saliency scores for each filter in your experiments, and how does it compare to those of related methods.
* Motivation for the choice of witness functions in numerical results is not well motivated, where do these choices of functions come from?
* Minor issue in Fisher discriminant eq. (2)... should be $\Sigma_q$ not $\Sigma_2$.
* Minor issue in algorithm description for class unlearning; the filter saliency scores should be class labeled $r^c_{\ell, j}$ and should be computed using equation (10) not equation (9).
* Applicability: method is currently restricted to image classification with convolutional networks.

**Questions:**

* Please fix the minor issues described above.
* Consider reporting computational performance of proposed method compared to baselines.
* Motivate the choice of witness functions in numerical results, and provide any ablations where applicable.
* Report model accuracies in all tables, and not just relative performance drops.
* Minor; consider adding figures to summarize performance and confidence intervals across all classes.
* Please consider elaborating and including numerical results indicating the impact of the number of classes on the ability to find class discriminative filters. These can be conducted on ImageNet by training models on subsets of the classes; e.g., 10 classes, 100 classes, 500 classes, 1000 classes.
* Please consider including experiments with a small vision transformer (e.g., ViT-Tiny) and discuss how the approach might extend beyond convolutional networks.

**Limitations:**

N.A.

---

> ### Author Rebuttal · Authors · 2024-08-07
>
> We thank the reviewer for the positive appraisal of our work, and especially the detailed feedback and insightful questions. We address the reviewers concerns below.
>
> ## Weaknesses
>
> **Numerical results both in the main paper and the appendix do not report the actual accuracies of any of the models. Instead, they only report relative drops in performance on forget and non-forget classes.**
>
> In this work, following standard practice (e.g. [3], [4]), we presented the accuracy drops. We present a table with the baseline accuracies, as requested by the reviewer in the Questions section. This can be seen as a reworking of Table 1, which will be repeated across tables in the paper.
>
> **The link provided for the ImageNet classifier in line 910 in the appendix does not work.**
>
> The referee refers to the Gdrive link from which we directly downloaded the VGG16 and ResNet50 models provided in the CHIP (our near baseline) GitHub page that later got updated by the authors of CHIP. We will update the link to the CHIP GitHub repo in the document.
>
> **Computational complexity of the approach compared to related works is not clear; specifically, what is the cost of computing the saliency scores for each filter in your experiments, and how does it compare to those of related methods.**
>
> Typically, complexity of the saliency computation is not provided in related work, such as CHIP [1] or TVSPrune [2]. In our work, we state the complexities of computing different types of witness functions in Appendix C.2, specifically Table 3.  For comparison, we also provide a table of wallclock times of computing saliencies *for all filters in the model* with our approach, as compared to CHIP and TVSPrune.
>
> **Imagenet (ResNet50)**
>
> |Method |Time  |
> |--|--|
> | CHIP[1] | >15 hours |
> | TVSPrune [2] | >15 hours  |
> | Ours | ~13 hours  |
>
> **CIFAR10 (VGG16)**
>
> |Method |Time  |
> |--|--|
> | CHIP [1] |  224 minutes |
> | TVSPrune [2] | 195 minutes|
> | Ours | 24mins |
>
> **Takeaway: Our proposed approach is computationally more inexpensive than common baselines and requires no loss function/backpropagation.**
>
> **Motivation for the choice of witness functions in numerical results is not well motivated, where do these choices of functions come from?**
>
> The choice of witness function is governed by relations to classical methods (i.e. Fisher discriminants/MPM), speed of saliency computation, and ease of estimation of moments. The witness function used in our pruning experiments (section 6.2) is described in lines 358-359. This enables us to recover the Fisher/MPM bounds stated in Theorems 1 and 2, and corollary 2, which are also easy to compute. Moreover, these witness functions are also in the spirit of connecting classical, discriminant-based classifiers to the TV distance/Bayes error rate, with which we can then derive novel bounds on the excess risk of those classifiers, as seen in section C.1.
> In section 6.1, we use the witness function stated in line 335, which is similar to the RBF Kernel or the moment generating function. A detailed study on the choice of witness functions is, however, the focus of our ongoing research.
>
> **Typos**
> We have amended the typos pointed out by the reviewer in the main document
>
> **Applicability: method is currently restricted to image classification with convolutional networks.**
>
> The results proposed in Theorems 1 and 2 can be applied whenever lower bounds on the total variation distance are required, and not just in the context of model editing. The techniques employed in deriving the key results can be applied to finding witness function-based lower bounds for other divergences as well.
>
> Moreover, when applied to model editing problems such as pruning and classwise unlearning, the proposed methods can be applied whenever we can access conditional data distributions.
>
> Our method applies to other model types as well. As requested by the reviewer, we will shortly present experiments showcasing the use of our methods on ViTs trained on CIFAR10 and CIFAR100.
>
> ## Questions:
>
> **Minor; consider adding figures to summarize performance and confidence intervals across all classes.**
>
> We will do so in the revised manuscript, and thank the reviewer for the suggestion.
>
> **Please consider elaborating and including numerical results indicating the impact of the number of classes on the ability to find class discriminative filters. These can be conducted on ImageNet by training models on subsets of the classes; e.g., 10 classes, 100 classes, 500 classes, 1000 classes.**
>
> We thank the reviewer for the interesting question. The discriminative filters hypothesis proposed in [2] was supported by several experiments, as well as those in this work (see Figs. 5, 6 of the manuscript). It is expected that discriminative filters are easy to identify if the width of the layers being investigated exceeds the number of classes. We see this for models trained on CIFAR100. The final layers of VGG16 (containing 512 filters) possess, on average, 103 (out of 3072) discriminative filters per class in the final 6 layers, whereas ResNet56 models trained on the same dataset possess on average 21 (in the final layer block). Experiments are ongoing for models trained on Imagenet and its subsets.
>
> **Takeaway: The ability to identify discriminative components depends on whether the number of classes exceeds the width of the network or not**.
>
> ### References
>
> [1] Sui et al. *CHIP: CHannel Independence-based Pruning for Compact Neural Networks*
>
> [2] Murti et al. *TVSPrune - Pruning Non-discriminative filters via Total Variation separability of intermediate representations without fine tuning*
>
> [3] Shah et al. *Decomposing and Editing Predictions by Modeling Model Computation*
>
> [4] Jia et al. *Model Sparsity Can Simplify Machine Unlearning*

---

> ### Author Response · Authors · 2024-08-11
> **Response to Reviewer**
>
> We thank you for your patience, and have provided the requested ViT experiments as a response to the General Rebuttal [**here**](https://openreview.net/forum?id=tuiqq1G8I5&noteId=MiXju1ozmG). We see that our proposed approaches for model editing (specifically structured pruning and model editing) are effective when applied on vision transformers as well.
>
> In our previous response, we have addressed the following points:
> - addressed various typos pointed out by the reviewer
> - discussed the motivation for the choice of witness function
> - discussed the applicability of work beyond CNNs, as well as the broad applicability of our key theoretical results
> - discussed how difficult it is to identify discriminative filters in different networks/different datasets
> - discussed the computational cost of our approach against common baselines and provided clock times for our method compared to baselines.
>
> We hope that we have adequately addressed your concerns, and are eager for further engagement. Given that the rebuttal period ends in 3 days, we would be grateful to know what we might do to improve your appraisal of our work.

---

> > ### Author Response · Authors · 2024-08-12
> > **Gentle Reminder about upcoming deadline**
> >
> > We would like to gently remind the reviewer that the author-reviewer discussion period ends in less than 48 hours. We hope we've addressed the concerns you raised in your detailed review of our work. We would really to engage with you further, and in particular, address any additional concerns you may have.

---

### Author Rebuttal · Authors · 2024-08-07

We thank the readers for their appreciation of our work. In particular, we thank reviewers for noting:
- The importance of the problem addressed by our work - that is, model editing with a view toward structured pruning and classwise unlearning/
- The efficient and simple nature of our solution to both the problems of structured pruning as well as classwise unlearning by way of model editing, and highlighting the hitherto unknown connection between them by using discriminative components.
- The rigorous derivations of the novel lower bounds on the Total Variation distance that require no assumptions on the distributions being compared (i.e. no Gaussianity assumption). We also take this opportunity to highlight three facts about our results. First, the bounds may be of general interest, and can be used whenever lower bounds on the TV distance are required. Second, the bounds reveal new connections between the TV distance and discriminant based classifiers (such as the minimax probability machine and the Fisher discriminant), which we use to derive novel excess risk bounds for these classifiers in Appendix C.1. Third, the techniques used to derive these bounds can be used to derive lower bounds on other divergences as well.

However, we address a common concerns raised by the reviewers.

**Comparison with Baselines for Model Unlearning**

As suggested by Reviewers, **H2n6**, and **crkh**, we compare our proposed method with recent additional baselines provided in [4]. Following [4],  we state the test accuracies on the forget and remain classes, averaged over all classes in the tables below . In the sequel, GA refers to Gradient Ascent, IU refers to influence unlearning (both as implemented in [4]), and l1-sparse refers to the approach as proposed in [4].

**CIFAR10 models**

VGG-16
|Method  |  Forget Class accuracy |Remain Class accuracy|
|--|--|--|
| **Ours** | 9.66% | 82.5% |
| [4], GA | 22.49% | 88.80% |
| [4], IU | 11.42% | 89.81% |

ResNet-20

|Method | Forget Class accuracy| Remain Class accuracy|
|--|--|--|
| **Ours** | 6.37% | 83.90% |
| [4], GA | 11.52% | 85.46% |
| [4], l1-sparse |  1.42% | 90.18%  |

**Takeaway: our model achieves superior forgetting compared to the baselines listed in [4] *notably without any fine-tuning of the model, and without utilizing the loss function in any way*, with minimal difference in remain accuracy.**


### References

[1] Sui et al, 2021. *CHIP: CHannel Independence-based Pruning for Compact Neural Networks*

[2] Murti et al, 2023. *TVSPrune - Pruning Non-discriminative filters via Total Variation separability of intermediate representations without fine tuning*

[3] Shah et al, 2024. *Decomposing and Editing Predictions by Modeling Model Computation*

[4] Jia et al, 2023. *Model Sparsity Can Simplify Machine Unlearning*

---

> ### Author Response · Authors · 2024-08-11
> **Additional Experimental Results - ViT pruning/unlearning**
>
> As requested by the reviewers, we present showcasing our pruning results on ViTs.
>
> We describe the model below:
>
> |  |  |
> |--|--|
> | Context Length | 65 |
> | Embedding dim. | 384 |
> | Transformer Encoders | 7 |
> | MLP layers to be pruned | 14 |
> | Total params. | 6.27m |
>
> We reduce the embedding dimension by pruning the MLP and Layernorm layers.
>
> We use the same architecture for CIFAR10 and CIFAR100 datasets, and we detail the test accuracies below.
>
> | Dataset | Test Acc. |
> |--|--|
> | CIFAR10 | 88.2% |
> | CIFAR100 | 69.5% |
>
>
> ### Pruning
> We prune the model by a fixed percentage without pruning the Attention heads - we prune the weights and biases of the MLP- and LayerNorm layers, without fine-tuning the model. Fully pruning attention heads is the focus of our ongoing research.
>
> **CIFAR10**
>
> | MLP Sparsity Ratio | Params Pruned | Test Acc (ours)| Relative Accuracy (w.r.t. unpruned model) |
> |--|--|--|--|
> | 30% | 1.24M | 82.7% | 94% |
> | 50% | 2.08M | 74.8% | 85% |
> | 70% | 3.12M  | 28.5% | 32% |
>
> **CIFAR100**
>
> | Sparsity Ratio | Params Pruned | Test Acc (ours)| Relative Drop |
> |--|--|--|--|--|
> | 30% | 1.24M | 60.3% | 87% |
> | 50% | 2.08M | 42.8% | 62%  |
> | 70% | 3.12M  | 5.2% | 8% |
>
>
> **Takeaway: our method achieves 50% sparsity on MLP layers while retaining 85% of the original accuracy on CIFAR10 *without any fine-tuning of the model*.**
>
> ### Class Unlearning
>
> We prune a small fraction of the rows of the MLP weight matrices and the layer norm weights.  We present the average Forget Class (FC) and Remain Class (RC) test accuracies over all classes.
>
> **CIFAR10**
> | Sparsity | Avg. FC acc. | Avg. RC acc. |
> |--|--|--|
> | 5.2%  | 16.5% | 66.3% |
>
> **CIFAR100**
> | Sparsity | Avg. FC acc. | Avg. RC acc. |
> |--|--|--|
> | 5.2%  | 13.1% | 54.2% |
>
> We also provide an additional experiment wherein we only prune the initial embedding layer and the LayerNorm layers, allowing us to achieve significant forgetting by editing fewer than 1% of parameters (approximately 40000 parameters).
>
> **CIFAR10**
> | Sparsity | Avg. FC acc. | Avg. RC acc. |
> |--|--|--|
> | 0.4%  | 7.3% | 40.6% |
>
>
> **Takeaway: Our method achieves significant reductions in forget class accuracy with minimal model edits *without fine-tuning*.**
>
> Note that our code base has been updated as well.

---

### Decision · Program_Chairs · 2024-09-25

**Decision:**

Accept (poster)

**Comment:**

The reviewers acknowledge the soundness and the technical novelty of the paper. The rebuttal has a positive impact on the final ratings and the reviewers scored 2 accepts, 1 weak accept, and 1 borderline reject. The ACs followed the reviewers' recommendation.